# B1 oligomerization regulates PML nuclear body biogenesis and leukemogenesis

Yuwen Li[1,2], Xiaodan Ma[1,2], Zhiming Chen[1,2], Haiyan Wu[1,2], Pengran Wang[1,2], Wenyu Wu[1], Nuo Cheng[1], Longhui Zeng[1], Hao Zhang[1], Xun Cai[1], Sai-Juan Chen[1], Zhu Chen[1] & Guoyu Meng [1]

ProMyelocyticLeukemia (PML) protein can polymerize into a mega-Dalton nuclear assembly of 0.1–2 μm in diameter. The mechanism of PML nuclear body biogenesis remains elusive. Here, PML$_{RBCC}$ is successfully purified. The gel filtration and ultracentrifugation analysis suggest a previously unrecognized sequential oligomerization mechanism via PML monomer, dimer, tetramer and N-mer. Consistently, PML B1-box structure (2.0 Å) and SAXS characterization reveal an unexpected networking by W157-, F158- and SD1-interfaces. Structure-based perturbations in these B1 interfaces not only impair oligomerization in vitro but also abolish PML sumoylation and nuclear body biogenesis in HeLa$^{Pml-/-}$ cell. More importantly, as demonstrated by in vivo study using transgenic mice, PML-RARα (PR) F158E precludes leukemogenesis. In addition, single cell RNA sequencing analysis shows that B1 oligomerization is an important regulator in PML-RARα-driven transactivation. Altogether, these results not only define a previously unrecognized B1-box oligomerization in PML, but also highlight oligomerization as an important factor in carcinogenesis.

[1] State Key Laboratory of Medical Genomics, Shanghai Institute of Hematology, Rui Jin Hospital affiliated to Shanghai Jiao Tong University School of Medicine, 197 Ruijin Er Road, Shanghai 200025, China. [2]These authors contributed equally: Yuwen Li, Xiaodan Ma, Zhiming Chen, Haiyan Wu, Pengran Wang. Correspondence and requests for materials should be addressed to G.M. (email: guoyumeng@shsmu.edu.cn)

The ProMyelocyticLeukemia (PML) protein, also known as TRIM19, is the core component of PML nuclear bodies (NBs) that are tightly associated with nuclear matrix. It has been reported that PML NB can play many instrumental roles in DNA-damage responses[1], apoptosis[2,3], cellular senescence[4,5], and angiogenesis, etc.[6]. It has been reported that PML NB can make direct handshakes with >120 different proteins, hence is recently recognized as PML interactosome[7]. Furthermore, via post-translational modifications like sumoylation, phosphorylation, acetylation, etc., PML can significantly influence its partners' fate and function in celleo[8–11]. All these have shaped PML NBs into the central hubs of cellular signaling pathways.

Until now, at least seven PML/TRIM19 isoforms have been reported[12]. Like most of other TRIM proteins[13], all of these variants share the conserved RBCC motif: an N-terminal RING, a B1-box (B1), a B2-box (B2), and a C-terminal coiled-coil (CC) (Fig. 1a). Previous studies have shown that the integrity of RBCC domain is essential for PML NB formation[12,14–16]. Furthermore, the PML sumoylation and SUMO-interacting-motif are also considered important for NB morphogenesis[16–18], but not essential[19,20]. Indeed, we have recently demonstrated that the PML oligomerization might occur prior to sumoylation[21]. However, due to the lack of success in obtaining purified PML or its RBCC fragment, our knowledge of PML NB biogenesis is extremely sparse. In this study, we report the largest RBCC fragment ever purified and the 2.0 Å crystal structure of B1 multimers. These results have prompted the in vitro and in vivo characterizations of PML and its oncogenic fusion PML-RARα (PR) in atomic detail.

## Results

**RBCC oligomerization.** Although early observation of PML NBs can date back to 1960s[22,23], the purification of PML protein remains a formidable challenge. This has significantly restricted the investigation and understanding of $PML_{RBCC}$-driven polymerization and nuclear body biogenesis (middle and right panels, Fig. 1a). In this study, a modified pET vector containing an N-terminal human SUMO tag was engineered. The presence of SUMO moiety had significantly enhanced the expression and folding of $PML_{1–256}$ that contained the RING, B1, B2, and a 20-residue CC. This allowed the subsequent purifications involving several chromatography steps (Fig. 1b). In order to investigate the RBCC oligomerization, the purified $PML_{1–256}$ was subjected to gel filtration (GF) analysis using a Superdex 200 column (Fig. 1c). Unexpectedly, $PML_{1–256}$ was eluted in three distinct peaks P1/P2/P3 with the estimated molecular weights ranging from 75 to 440 kDa, suggestive of RBCC dynamic in solution. To investigate the RBCC oligomerization further, ultracentrifugation (UC) characterization was used to analyze these GF peaks (Fig. 1d). The P3 peak, corresponding to the GF eluant with the lowest molecular weight, clearly displayed a 1/2/4-mer mixture (Fig. 1d). Consistently, the same oligomeric dynamic was observed in the GF peaks/species P1 and P2. More supportively, the high order assembly (N-mer), which was not observed in P3, started to appear and build-up in P1 and P2 (Fig. 1d), leading to hypothesis of a previously unrecognized, sequential oligomerization mechanism via 1/2/4/N-mers.

**B1-box multimers.** In line with the GF/UC analysis above, the structure of B1-box was determined to 2.0 Å resolution, unveiling an unexpected oligomerization in PML (Figs. 2, 3, and Supplementary Figs. 1 and 2). In the B1-box monomer, the residues 120–127 fold into a loop-like subdomain 1 (SD1), which is ~12 Å away from the B1 core. In comparison, the residues 128–167, coordinated by two Zn ions, fold into a compact global domain,

termed subdomain 2 (SD2). The relative orientation between SD1 and SD2 gives rise to a tea-cup-shape architecture, which is also observed in the NMR B1 monomer (Fig. 2a and Supplementary Fig. 2a). The subdomain 2 is a typical B-box-type Zn finger fold with a conserved β1–β2–α1–β3 arrangement and Zn-binding residues (Fig. 2b). The three strands, which are located in between the α1 helix and SD1, form an anti-parallel β-sheet. The α1 helix and β-sheet are coordinated with two Zn ions, residues C129/C132/C148/C151 (Zn1) and C140/C143/H155/H161 (Zn2). Above the Zn2 site is K160 (Fig. 2a), a critical sumoylation site for leukemia development[24]. The long lysyl side-chain and the amine head are fully exposed to solvent. This spatial location is consistent with its role as a SUMO acceptor. Next to K160 are two bulky residues, W157 and F158. The side chains of W157 and F158 are forming a 90° angle to each other, which is important for B1-oligomerization (Fig. 2c, d).

It has been reported that TRIM B1/2-box could mediate multimerization (Fig. 2b)[25–28]. This is also the case for PML B1-box. The W157, F158, and SD1 could facilitate the formation of PML B1 dimers (Fig. 2c–e). In W157-dimer, the inter-molecular engagement is mainly mediated by two sets of hydrophobic interactions between W157 and F152/F138 (Fig. 2c). In F158-dimer, the bulky hydrophobic side-chains are located in the heart of the dimeric interface (Fig. 2d), reminiscent of L73–L73 handshake in PML RING dimerization[21]. Interestingly, the Trp-Trp, Phe-Phe, and Leu-Leu interactions are often observed in TRIM multimerization (Supplementary Fig. 2b), highlighting the conserved α1 helices in B-boxes as an oligomeric hot spot. In addition to the F158–F158 interaction, the hydrogen bonds between W157–Q145 (2.7 Å) and R131–E153 (3.2 Å) can facilitate the B1–B1 interaction (Fig. 2d). In SD1-dimer, the N-terminal I122 and V123 residues could undergo a loop-to-strand transition to seal two B1s together via β-strand augmentation (Fig. 2e).

As revealed by crystal packing, these dimeric interfaces are the foundation of B1-driven higher order assembly (Fig. 3). The intermittent engagement between W157- and F158-interfaces enables the formation of a "brick"-like tetramer (Fig. 3a and Supplementary Fig. 2c). This, in turn, gives rise to two interesting positions of K160s and SD1s. In the α1-mediated W-shape coiled-coil, the K160 residues are consistently placed at the bottom of the B1 tetramer, giving rise to a highly concentrated Lys patch/lining. Supportively, only this Lys position, but not K65 and K490, allows poly-SUMO reaction, consistent with the K160 lining in W157/F158-tetramer (Supplementary Fig. 2c). In addition, this tetramer is flanked with four SD1 subdomains that are ready for further β-augmentation (Fig. 3b). As a result, remarkable networking mediated by B1-box could be envisaged (Fig. 3c).

**Functional characterizations of B1-driven oligomerization.** In order to check PML B1-box multimerization in solution, small-angle X-ray scattering (SAXS) was used. In this experiment, the atomic structures of the PML B1-box monomer, dimers, tetramer, and N-mer reported here were subjected to crystal fitting using the OLIGOMER algorism implemented in CRYSOL[29]. The SAXS characterization is in good agreement with the crystallographic observation ($\chi^2 = 1.21$), suggestive of the B1 oligomerization in solution (Fig. 4a). Furthermore, when the W157- and F158-interfaces were perturbed, the B1 multimerization (but not the overall fold, Supplementary Fig. 3a, b) was severely impaired, precluding B1 tetramer and higher-order assemblies (Fig. 4b–d). For further cross-validation, an oligomerization-dependent crosslinking assay was performed (Fig. 4e, f). When an increased concentration of cross-linker GA were incubated with

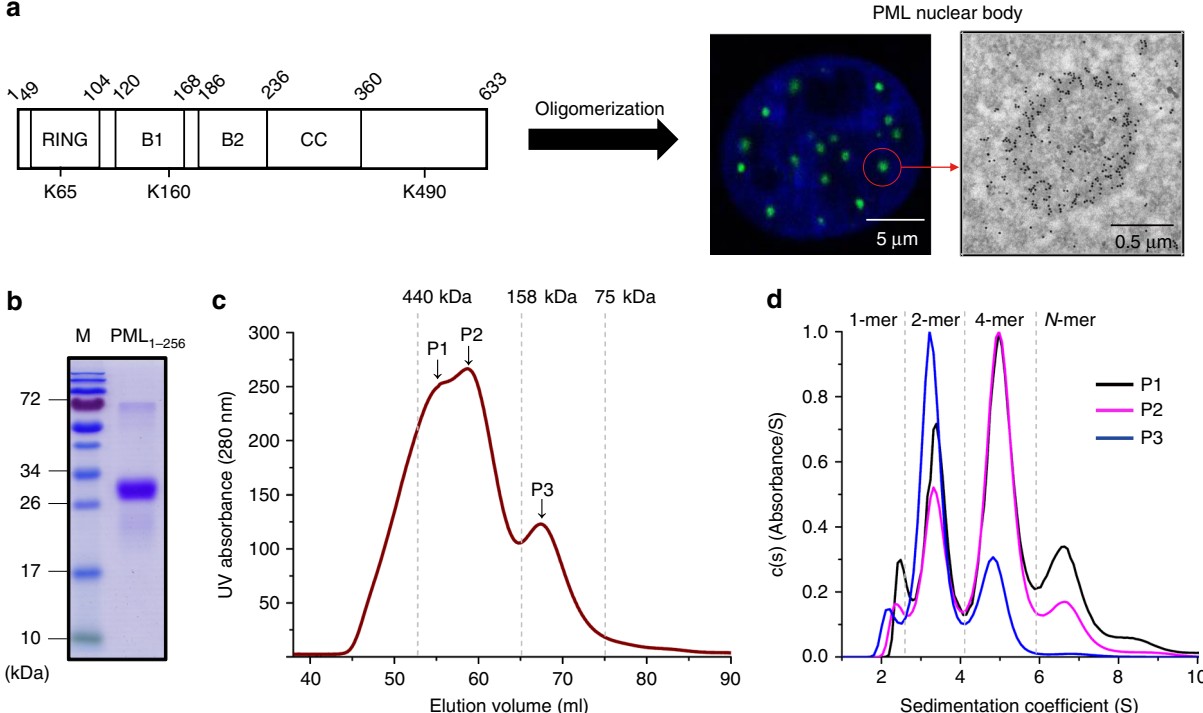

**Fig. 1** Gel filtration (GF) and ultracentrifugation (UC) characterization of PML RBCC domain. **a** Oligomerization is essential to PML nuclear body (NB) biogenesis. Left panel, the domain arrangement of PML isoform IV. The starting and ending residues of the RING, B1-box, B2-box, and coiled-coil (CC) domains are annotated above the schematic diagram. The sumoylation sites K65/K160/K490 are also indicated. Middle panel, HeLa$^{Pml-/-}$ cells that expressing wild type GFP-PML (green). Scar bar is 5 µm. The nucleus is stained with Hoechst (blue). Right panel, the immuno-gold electron microscopy visualization of a single PML NB (circled, middle panel), scar bar is 0.5 µm. The black dots in this micrograph are PML specific immune-gold particles. **b** Purification of recombinant PML RBCC$_{1-256}$ as monitored by SDS-PAGE. The theoretical molecular weight (MW) of PML$_{1-256}$ is 28 kDa. **c** Gel filtration analysis of PML$_{1-256}$. The elution peaks are designated as P1–P3, respectively. Source data are provided as a Source Data file. The Superdex 200 column used in this study was calibrated using the standard molecular marker kit (GE Healthcare). The reference molecular weights are indicated. **d** Analytical ultracentrifugation analysis of PML$_{1-256}$ gel filtration eluants/peaks. The sedimentation coefficients of Peaks 1/2/3 are colored in black, magenta and blue, respectively. From left to right, the molecular masses of the UC peaks are 34, 54, 99, and 162 kDa, suggesting the assemblies of PML$_{1-256}$ monomer, dimer, tetramer and much higher order polymers (termed N-mer in this report). Source data are provided as a Source Data file

the wild-type (WT) PML B1 and PML$_{1-256}$, we could detect a growing accumulation of B1 and RBCC polymers in sodium dodecylsulphate polyacrylamide gel electrophoresis (SDS-PAGE) and western blots. In contrast, W157E, F158E, and I122P/V123P mutants displayed marked resistance to the GA cross-linking when compared to the WT proteins (Fig. 4e, f). Of note, I122P/V123P displayed relative minor disruption than those observed in W157E and F158E (Fig. 4e, f). This seems to suggest that SD1-interface might contribute to the later B1 oligomerization step after W157/F158-tetramerization (see more supportive results below).

Besides, the B1 oligomerization was also validated by GF and analytical UC (Fig. 5a, b). As mentioned above, WT PML$_{1-256}$ was eluted with three peaks containing 1/2/4/N-mers. In comparison, W157E and F158E significantly altered the PML$_{1-256}$ elution profile (Fig. 5a). The monomeric peak (P4), which was not observed in WT, was now visible in mutants. The same results were obtained by analytical UC analysis (Fig. 5b). Similar disruptive impact, but at the later stage of PML polymerization, could be observed for I122P/V123P (Fig. 5b). This was further supported by the mammalian two-hybrid assay using the full-length PML isoform IV protein (Fig. 5c). In this assay, we observed similar disruptive impacts by W157E, F158E, and I122P/V123P. The mutations all consistently impaired the PML self-interaction (Fig. 5c). More importantly, the PML isoform IV mutants W157E, F158E, and I122P/V123P, although expressed normally in HeLa$^{Pml-/-}$ cells, precluded NB formation

and PML sumoylation at basal level (Fig. 5d–f and Supplementary Fig. 4a). Consistently, similar results could be obtained in PML isoform I (Supplementary Fig. 5a, b). Complementary to these results, we also observed the same results in W157K, W157A, F158K, F158A, I122P, V123P, and Δ120–124 (Supplementary Fig. 4b, c). In marked contrast, the mutations of D137A, S135A/D137A, and A149E that targeted the NMR B1 dimeric interface had little impact on NB formation (Supplementary Fig. 4d, e), urging caution in the interpretation of NMR B1 dimer[26].

It has been reported that partners recruitment is also important for NB formation and function[16,19]. We next want to know how the loss of RBCC oligomerization might affect the PML recruitment activity. Using immuno co-localization and mammalian two-hybrid assays, we examined the interactions between PML and DAXX, SUMO2/3 in HeLa$^{Pml-/-}$ cells (Fig. 6a, b and Supplementary Fig. 6a, b). In the co-localization assay, the sumoylation mutation K160R was used as control. As expected, DAXX and SUMOs could colocalize with WT NBs, but not with the K160R mutant, reiterating the importance of sumoylation in PML partners recruitment[19]. Consistently, the B1 mutants, which also abolished PML auto-sumoylation at basal level (Fig. 5e), displayed reduced DAXX/SUMOs recruitment activity (Fig. 6a, b and Supplementary Fig. 6a, b). Of note, I122P/V123P, which targeted the possible later step in B1 polymerization, could still form NBs with the abnormal bigger size, and hence displayed NB-enhanced partners recruitment in the co-localization assay. In addition, when the B1 mutants were subjected to arsenic trioxide

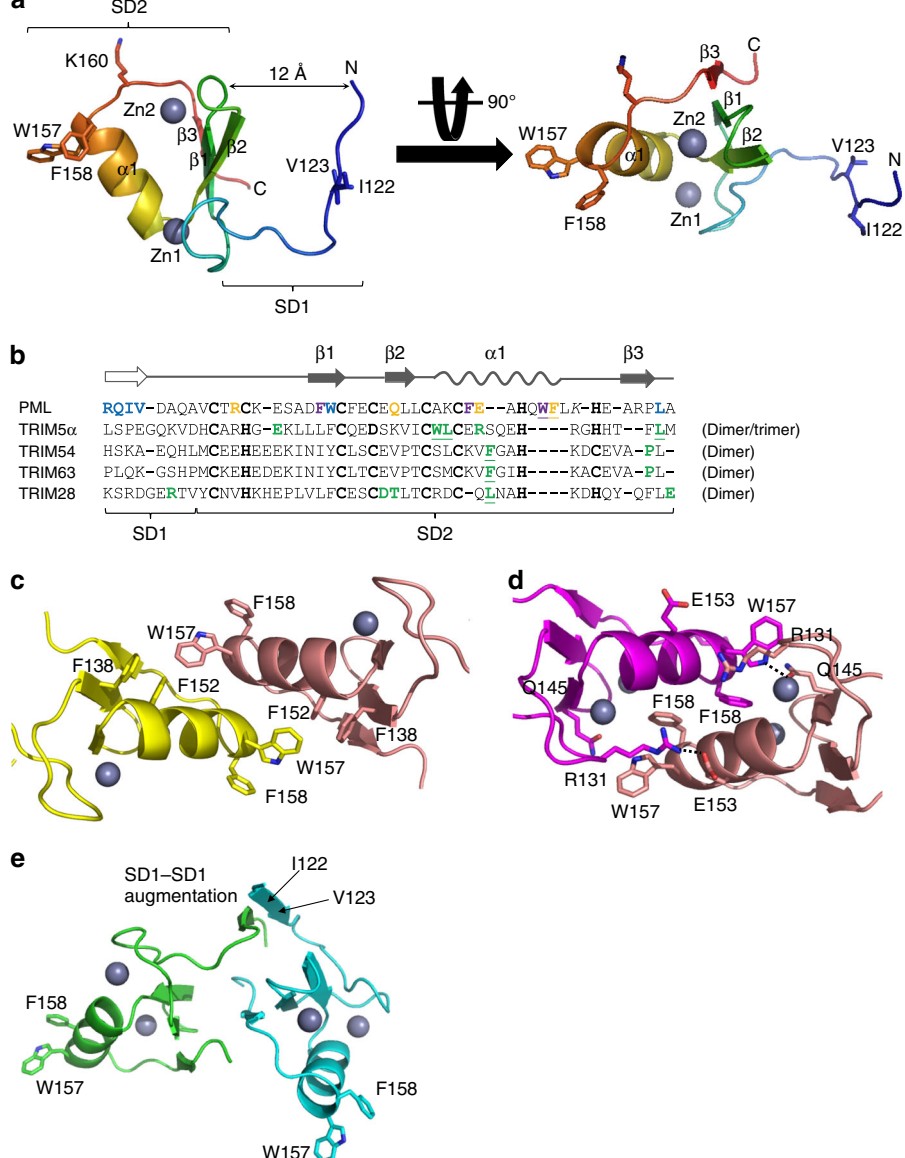

**Fig. 2** Crystal structure of PML B1-box multimers. **a** B1-box monomer. The crystallographic monomer containing residues 120–167 is shown in cartoon representation. From N- to C-terminus, the structure is colored using the rainbow scheme implemented in Pymol. I122, V123, W157, F158, and K160 are shown in stick representation. The Zn1 and Zn2 ions are shown in sphere representation. The subdomain 1 (SD1) and subdomain 2 (SD2) are bracketed. **b** Sequence alignment of B-box driven oligomerization. The secondary structures of PML B1-box are shown on top of the sequences. An empty arrow is used to highlight the loop-to-strand conversion during SD1-augmentation (see below, Fig. 2e). The highly conserved Zn-binding residues are highlighted with bold font. The residues, which are involved in W157-, F158-, and SD1-dimerizations, are colored in purple, yellow, and blue, respectively. The residues, which are observed in other TRIM oligomerizations, are colored in green. In particular, the conserved hydrophobic interactions in α1 helices (Supplementary Fig. 2b) are underlined. **c** W157-dimer. The residues W157, F138, and F152 in the dimeric cleft are shown in stick representations. **d** F158-dimer. The auxiliary hydrogen bonds surrounding F158–F158 are shown in dashed lines. **e** SD1-interface. The residues I122 and V123 that lie in the heart of SD1-augmentation are highlighted with arrows

(ATO) treatment, we could observe NBs re-appearance, accompanied with NB-enhanced partners recruitment (Fig. 6c and Supplementary Fig. 6c). Similar partners recruitment and ATO rescue results could also be repeated in PML isoform I (Supplementary Fig. 5c, d). Altogether, these results highlight the idea that the RBCC oligomerization is likely to be the first step in PML NB biogenesis, preceding PML sumoylation and partners recruitment.

Interestingly, when various PML mutants were subjected to ATO treatment, we observed a striking rescue difference between B1 and RING oligomerizations. Unlike L73E in RING[21], the

damage caused by F158E or W157E/F158E in B1-box could be rescued by arsenic treatment (Fig. 7). After short exposure of As$_2$O$_3$, F158E and W157E/F158E could reform NBs (Fig. 7a, b). As a result, the B1 mutants recovered some levels of auto-sumoylation (Fig. 7c). In marked contrast, no rescue effect could be observed in RING mutant L73E (Fig. 7). The results not only supported our previous claim that PML–UBC9 interaction might occur in L73–RING dimer[21], but also suggested a cooperative mechanism among RBCC, in which the RING tetramerization step might precede B1 polymerization in PML nuclear body biogenesis.

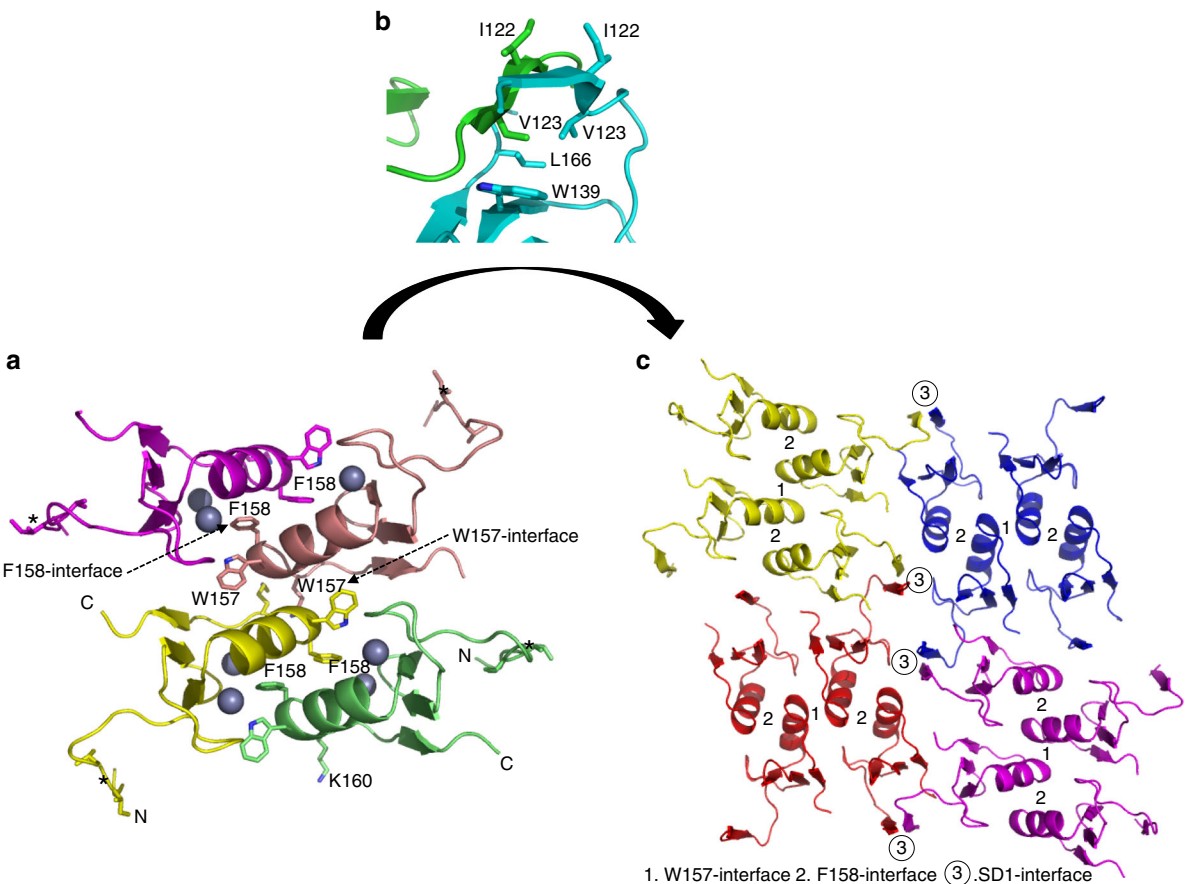

**Fig. 3** Crystal packing reveals an oligomerization network by PML B1-box. **a** A possible B1 tetramer. B1 monomers are colored in magenta, brown, yellow, and green, respectively. The W157- and F158-interfaces are arrowed. The SD1-interface is highlighted with "*". Residues I122, V123, W157, F158, and K160 are shown in stick representation. **b** Enlarged view of SD1–SD1 dimer. The residues that facilitate the formation of this dimerization are shown in stick representation. **c** B1-box polymers (N-mer). The W157/F158-tetramers are colored in yellow, blue, purple, and red, respectively. B1 networking is mediated by W157-, F158-, and SD1 interfaces

**B1-box multimerization in PR**. Concerning PR, we wanted to know whether the B1-box oligomerization is an important regulator in PR-driven transactivation and leukemogenesis. Firstly, the presence of B1 oligomerization in PR was confirmed by speckle formation assay[30], in which W157E, F158E, and I122P/V123P consistently disrupted the PR nuclear dots (Supplementary Fig. 7a). Secondly, in vitro luciferase assay[31] was used to monitor the impact of the multimerization upon PR transcription activity (Supplementary Fig. 7b). The results were also coherent, supportive of B1 multimerization in oncogenic fusion. Thirdly, using PR or PR F158E transgenic mice, we also observed a strict correlation between B1-oligomerization and leukemogenesis (Fig. 8a and Supplementary Fig. 7c–e). Over one and half year, most PR mice ($n = 8$) died from APL. In marked contrast, none of PR F158E mice developed leukemia (Fig. 8a). In order to get insight into the leukemogenic transactivation program triggered by oligomerization, the PR, PR F158E transgenic mice, and the normal FVB/N mice (termed WT) were subjected to single cell RNA sequencing analysis (Fig. 8b, c and Supplementary Table 1). According to the gene-expression profiling, the bone marrow cells from WT (3816), PR (4806), and PR F158E (3874) were classified into 15 different clusters that could be recognized as granulocyte, erythrocyte, monocyte, stem cell, B cell, and dendritic cell (Fig. 8b). The t-distributed stochastic neighbor embedding plots of the key markers used in the classification of different cell types are shown in the Supplementary Fig. 8a. The cell type distributions for PR were 50.4%, 39%, 7%, 1.9%, 1.2%, and 0.5% for

granulocyte, erythrocyte, monocyte, stem cell, B cell, and dendritic cell, respectively. In comparison, PR F158E displayed a marked difference in cell differentiation (Fig. 8c). The cell type distributions for PR F158E were 26%, 63%, 3.9%, 4.5%, 0.9%, and 1.7% for granulocyte, erythrocyte, monocyte, stem cell, B cell, and dendritic cell, respectively (Supplementary Fig. 8b, c). In addition, the differential expression analysis of PR and PR F158E datasets with stringent threshold values of log2[fold change] > 1.5 and the adjusted $p$ value/FDR < 0.5 helped to uncover a new set of PR-oligomerization target genes (Fig. 9a, b and Supplementary Table 2). Consistently, Trib3 and Atf5, which were upregulated by PR and recently implicated in APL[32], were clearly downregulated by PR F158E (Fig. 9a, b). Altogether, these results prompt the new thinking of PR, in which oligomerization might be a crucial regulator for leukemia development (Fig. 9c).

## Discussion

The B-box domain has been widely observed in nature[33,34]. Using the Dali server, an invariant PML B1-box fold could be observed, albeit with low sequence identity. They are ADP-ribosylation factor 1 (RMSD 1.5 Å and sequence identity 18%), ubiquitin carboxyl-terminal hydrolase 8 (RMSD 1.8 Å and sequence identity 26%), ubiquitin carboxyl-terminal hydrolase 5 (RMSD 1.8 Å and sequence identity 13%), ubiquitin E3 ligase Bre1a (RMSD 2.3 Å and sequence identity 18%), ubiquitin E3 ligase RNF123 (RMSD, 2.2 Å and sequence identity, 16%), DNA gyrase subunit A (RMSD 1.8 Å and sequence identity, 9%). This implies that

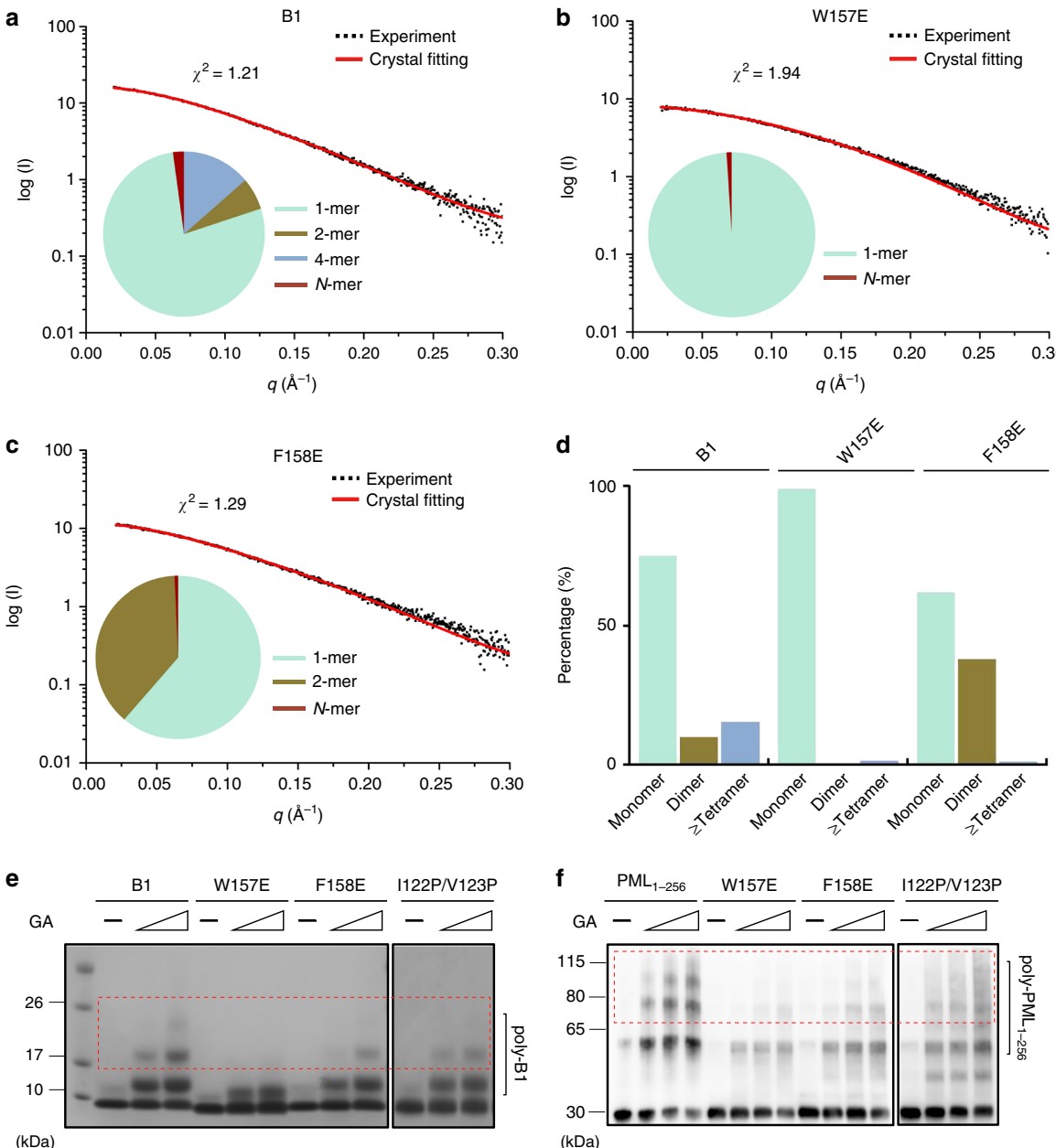

**Fig. 4** SAXS and cross-linking assays of PML B1-box oligomerization. **a–c** SAXS analysis of WT B1 (**a**), B1 W157E (**b**), and B1 F158E (**c**). The experimental data are shown in black dots. The theoretical scattering pattern derived from PML B1 multimers is shown in red line. Bottom-left panels of each figure, the oligomeric distribution/dynamic in solution. Source data are provided as a Source Data file. **d** Statistical summary of SAXS experiments. **e**, **f** Oligomerization-dependent crosslinking assays. In **e**, PML B1 and mutants were cross-linked by an increased concentration of glutaraldehyde (GA). In **f**, PML$_{1-256}$ and mutants were cross-linked by an increased concentration of glutaraldehyde (GA). The poly-B1 and poly-PML$_{1-256}$ are highlighted with brackets and red boxes. Source data are provided as a Source Data file

B-box might serve as a power scaffold during protein evolution. The variations stemming from a conserved B-box fold might be important to protein diversities, including its enzymatic activities and assembly/oligomerization. Indeed, the B-box, which is often associated with RING and coiled-coil (CC) in TRIMs, has been reported to facilitate the assemblies of different oligomerization[25–28]. Different hydrophobic decorations in the conserved α1 helices lead to various oligomerizations (Supplementary Fig. 2b). Notably, the functional dimer-to-trimer switch regulated by Trp–Trp interaction (Supplementary Fig. 2b) is the key structural determent for TRIM5α to polymerize into a mega-Dalton hexagonal network that is required for HIV-1 virus recognition[28].

In this study, we also report a previously unrecognized B1-box polymerization ($n > 4$) that is essential for PML/TRIM19 nuclear body assembly. The oligomeric determinants such as W157, F158, and SD1-loop are uniquely observed in PMLs, but not in other TRIMs (Supplementary Figs. 1 and 2). Through W157-, F158-, and SD1-interfaces, PML B1-box can form homo-dimers, tetramer and much higher order polymer (Figs. 2 and 3). The crystallographic observation is further supported by in vitro characterizations (Figs. 4 and 5 and Supplementary Fig. 3). More importantly, in the context of RBCC and full-length PML, perturbations in the oligomeric interfaces significantly impaired PML self-interaction and precluded NB biogenesis, auto-sumoylation and partners recruitment (Figs. 5–7 and

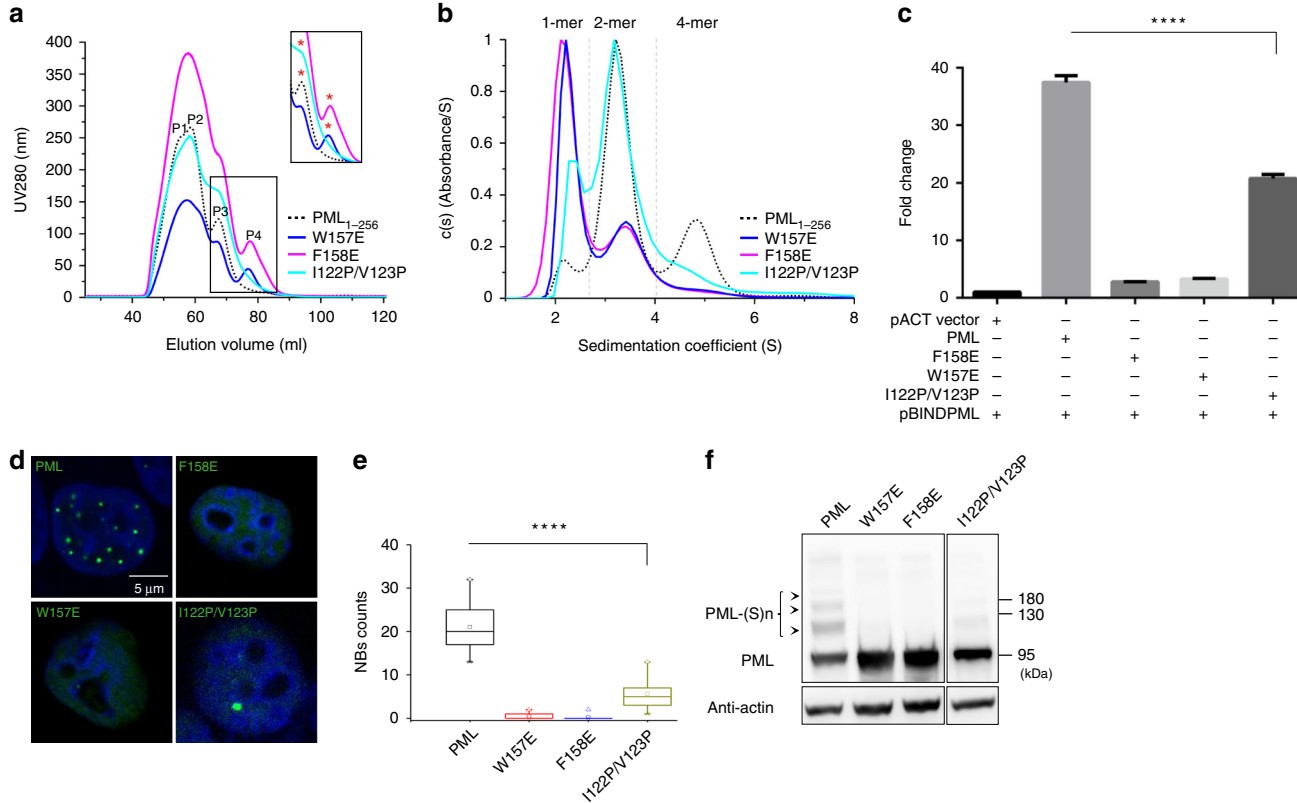

**Fig. 5** More in vitro and in vivo analyses of RBCC oligomerization and PML NB biogenesis. **a** Gel filtration characterization of WT and mutant PML$_{1-256}$. WT, dashed line. W157E, blue line. F158E, magenta line. I122P/V123P, cyan line. The elution P3/P4 peaks that were subjected to analytical ultracentrifugation analysis in (**b**) are highlighted with "*". Source data are provided as a Source Data file. **b** Analytical ultracentrifugation assay. The WT and mutants are colored using the same scheme as (**a**). The monomeric, dimeric, and tetrameric assemblies are separated with dashed lines. Source data are provided as a Source Data file. **c** Mammalian two-hybrid assay. The relative luciferase activities (RLU) were used to monitor the PML self-interaction. The interactions between WT and different mutants are all normalized against the pACT vector:pBIND-PML interaction (=1). Values are means ± S.E. ****$p < 0.0001$ (derived from student $t$ test) are used to show statistical significance between WT and mutants. All experiments have been done >3 independent replicates. Source data are provided as a Source Data file. **d** Immuno-fluorescence of HeLa$^{Pml-/-}$ cells that contained EGFP-PML, EGFP-PML$_{W157E}$, EGFP-PML$_{F158E}$, and EGFP-PML$_{I122P/V123P}$. Scar bar is 5 μm. **e** Statistical analysis of PML nuclear body formation. All experiments have been done in 6 independent replicates, NB counts were calculated from ≥15 nuclei. Values are means ± S.E. ****$p < 0.0001$. All the experiments/results displayed in the main figure are done with PML isoform IV. Source data are provided as a Source Data file. **f** PML sumoylation at basal level. The auto-sumoylation of HeLa$^{Pml-/-}$ stably expressing B1 mutants was monitored by western blot using antibody against HA tag. Source data are provided as a Source Data file

Supplementary Figs. 4–7). However, it is worthy to point out that the B1-mutations-alone cannot completely stop RBCC oligomerization (Fig. 5a, b), reiterating the importances/contributions of RING and CC oligomerization. In the previous studies, we know ATO can facilitate PML NB formation[35]. Here, as shown in Fig. 7, we observed different ATO responses in RING and B1 mutants. The lack of ATO rescue effect in L73E mutant supports a cooperative oligomerization mechanism between RING, B1/2-box and CC domains. However, based on current data, it is not yet clear how RING, B1/2-box and CC might cooperate, and via what mechanism a 2D B1-network can fold into a 3D PML speckle. The recent studies showed that the combination of B-box trimerization and CC trans-dimerization is a reasonable strategy utilized by TRIM5α to reach a mega-Dalton hexagonal network[28]. This has promoted the idea/speculation that the contribution of RING and CC oligomerization might also be critical to enable the final 3D assembly in PML/TRIM19.

In previous study, we have shown that the L73E mutation, which inhibits PML RING tetramerization, is essential for leukemogenesis[21]. Using single-cell RNA sequencing technology, we can detect significantly altered PR-driven transactivation and differentiation in the PR F158E mouse. More consistently, the B1 single mutation that targets the PR oligomerization again

precludes APL development in vivo (Figs. 8 and 9). This is further supported by the observation that PML coiled–coil interaction is also important for differentiation arrest and transformation in vivo[36]. All these results are coherent, highlighting RBCC oligomerization as an important regulator in leukemogenesis. Actually, the oligomerization theme might not be restricted to PR and APL. In oncogenic fusions NPM-RARα and Stat5b-RARα, oligomerization is thought to be instrumental to cell immortalization and diseases development[37,38]. In proto-oncogenic proteins erbB-2, MET and FGFR, the enhanced and gain-of-carcinogenic effects by multimerization are also observed in tumor growth and cancer progression[39–42]. Like the W157- and F158-interfaces observed in PML B1 oligomerization, the cancer-related multimerizations are often mediated/stabilized by the bulky hydrophobic residues such as Trp, Phe, Leu, Ile, and Val[43–47]. Indeed, leukemogenic oligomerizations are widely observed in BCR/ABL, AML1/ETO, AML1/MTG16, TEL/AML1, CBF$_β$/SMMHC, MLL/GAS7, MLL/AF1p, MLL/GEPRIN, and PAX5/PML, etc.[48–52]. In light of the single cell RNA sequencing results presented here, it might be reasonable to consider oligomerization as an important leukemogenic regulator in these drivers. One of the important research avenues in the future is to investigate whether/how the oncogenic oligomerization might influence the maintenance of the

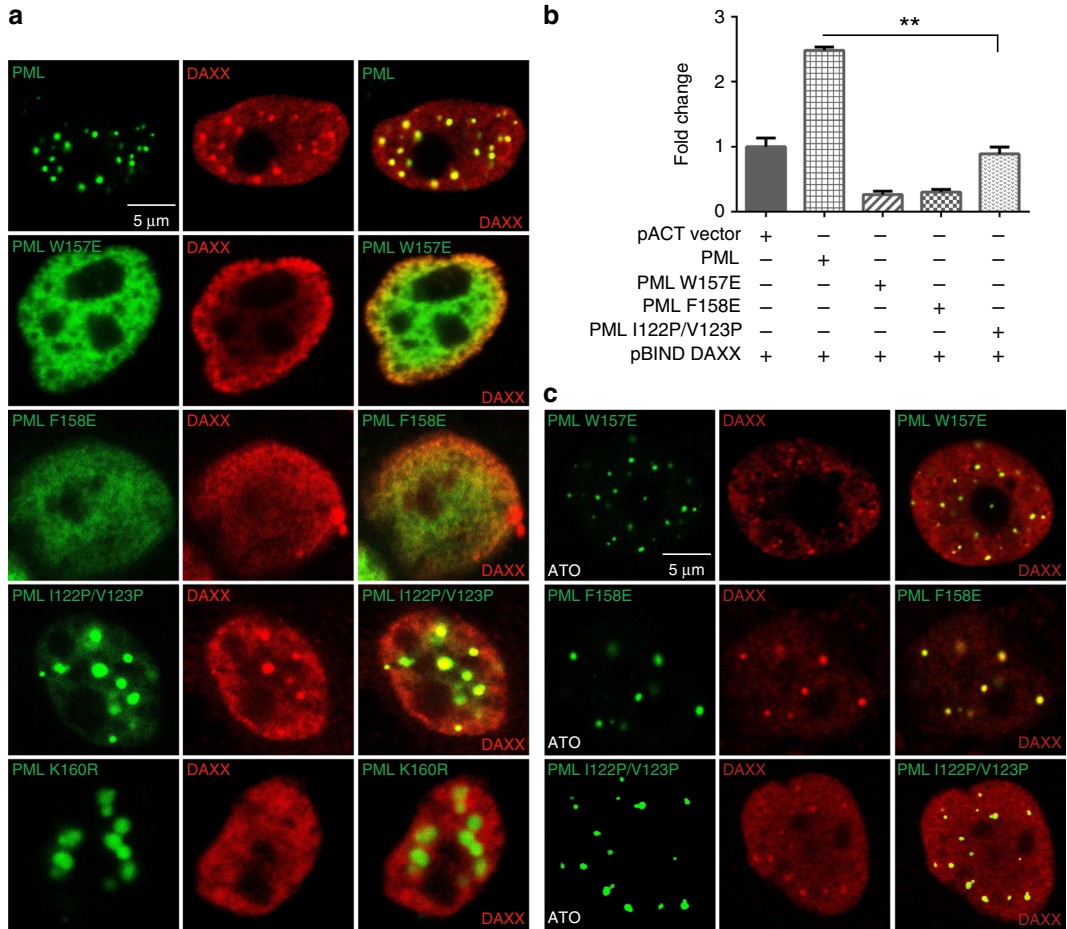

**Fig. 6** Immuno co-localizations of PML and partner proteins. **a** Immuno co-localization between PML isoform IV and DAXX. The expressions of GFP-PML and mutants were visualized by immunofluorescence. Scar bar is 5 μm. The expressions of DAXX were detected using antibody against FLAG tag. In addition to the newly reported B1 mutations, the K160R mutant, which impairs the PML sumoylation[24], was also monitored. **b** Mammalian two-hybrid characterization between PML isoform IV and DAXX. The interactions between WT/mutants and DAXX are all normalized against the pACT vector: pBIND-DAXX interaction (=1). Values are means ± S.E. **$p$ < 0.01. Source data are provided as a Source Data file. **c** NB-enhanced partners recruitment by arsenic trioxide (ATO). HeLa$^{Pml-/-}$ cells expressing B1 mutants were exposed to 2 μM As$_2$O$_3$ before visualization by immunofluorescence

leukemia-initiating cancer cells[53]. If this is the case, the oncogenic oligomerization might represent a valuable target for future therapeutic exploration beyond APL[54,55].

## Methods

**DNA and protein preparation.** PML B1-box consisting of residues 120–168 was constructed into a pET 32a vector with a cleavable His tag on the N-terminus. For PML$_{1–256}$, a modified pET 15b was used. The cDNA encoding human SUMO moiety was engineered into pET 15b using restriction *NcoI* site via homologous recombination. The RBCC fragment encoding residues 1–256 was then ligated into the modified pET 15b vector using *NdeI* and *XhoI* sites. The resulting plasmid encoded a cleavable His tag in between SUMO and PML$_{1–256}$. All the PML and PR mutants reported here were engineered using the Quick Change II site-directed mutagenesis kit (QIAGEN). The primers used in this study were reported in Supplementary Tables 3 and 4.

The PML B1-box or PML$_{1–256}$ was transformed into *Escherichia coli* BL21 (DE3) (Sangon) for protein expression. The cells were grown in LB Broth supplemented with 100 μg/ml ampicillin at 37 °C. When OD$_{600}$ reached ~0.6, the cells were induced with 300 μM IPTG (Sangon) and 300 μM ZnCl$_2$ (Sangon) for >12 h. The cells were harvested by centrifugation (4700$g$, 20 min). The cells were then resuspended in a buffer containing 20 mM Tris, pH 8.0, 100 mM NaCl before a French press treatment (JNBIO). The clear lysate was obtained by centrifugation (46,000$g$, 1 h). The target protein was purified through a nickel column (His-Trap, GE Healthcare), followed by thrombin (Invitrogen) digestion at 20 °C. The TRX-His or SUMO-His tags were removed by His-Trap columns. The PML B1-box and RBCC$_{1–256}$ protein samples were further purified by an anion exchange column (Q, GE Healthcare), in which the buffer A and B were 20 mM Tris, pH 8.0, 20 mM NaCl and 20 mM Tris, pH 8.0, 1 M NaCl, respectively. The eluant containing

the target proteins was pooled and further purified by the hydrophobic chromatography (Phenyl, GE Healthcare), in which the buffer A and B were 20 mM Tris, pH 8.0, 20 mM NaCl, 1 M (NH4)$_2$SO$_4$ and 20 mM Tris, pH 8.0, 20 mM NaCl, respectively. In the final GF step, the columns S100 and Superdex 200 (GE Healthcare) were selected for PML B1-box and PML$_{1–256}$, respectively. The GF buffer was 20 mM Tris, pH 8.0, 100 mM NaCl.

**Crystallization and structural determination.** PML B1-box crystals were obtained by the vapor diffusion technique. The purified protein (38 mg/ml) was mixed at a 1:1 (v/v) ratio with the buffer containing 30% polyethylene glycol 6000, 1 M lithium chloride and 100 mM sodium acetate. The crystals were stabilized in a mixture of paraffin and paratone-N (Hampton Research), followed by X-ray examination at 100 K. Diffraction data were collected in Beamline station BL17U at Shanghai Synchrotron Radiation Facility (SSRF, Shanghai, China). Zn anomalous dispersion at 1.2824 Å was used for data collection and subsequent phasing. The diffraction data were processed, integrated and scaled using MOSFLM/SCALA[56]. The statistics of the data collection are shown in Table 1.

Single wavelength dispersion algorism in CRANK2, together with data in 20 and 2.0 Å, were used to obtain the first set of phases for PML B1-box. Eight Zn positions were successfully determined and the resulting electron density map was improved by solvent flattening program SOLOMON[56] and auto-tracing program ARP/warp[57]. This procured an excellent σ$_A$-weighted *2Fo–Fc* map for manual model building using COOT[56] and structural refinement using REFMAC5[58] and Phenix.refine[59]. The final model in ASU contains 196 residues, 8 Zn ions, and 82 water molecules. Ramachandran statistics by PROCHECK[60] showed that 97.4% of the atoms are in the most favored region, and 2.6% are in the allowed regions. The detailed structure refinement statistics are reported in Table 1. The PML B1-box structure has been deposited into the Protein Database Bank with the entry code of 6IMQ.

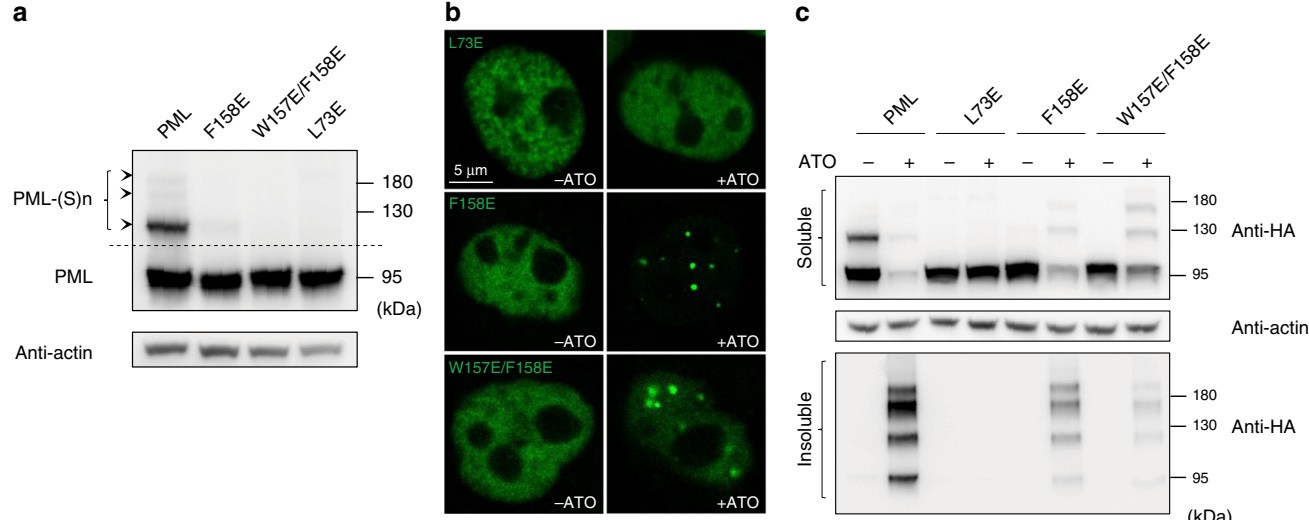

**Fig. 7** B1-oligomerization might occur after RING tetramerization. **a** Auto-sumoylation of PML RING and B1 mutants at basal level. The auto-sumoylation of HeLa$^{Pml-/-}$ expressing B1 and RING mutants was monitored by western blot using antibody against HA tag. Source data are provided as a Source Data file. **b** NB formation rescued by ATO. Left columns: HeLa$^{Pml-/-}$ cells without arsenic treatment. Right columns: One-hour exposure of 2 μM As$_2$O$_3$ before visualization. **c** PML sumoylation rescued by ATO. The WT and mutant cells, which had been treated with 2 μM As$_2$O$_3$, were disrupted by RIPA buffer. The PML sumoylation that precipitated the soluble PML into insoluble matrix was analyzed by western blot using antibody against HA tag. Source data are provided as a Source Data file

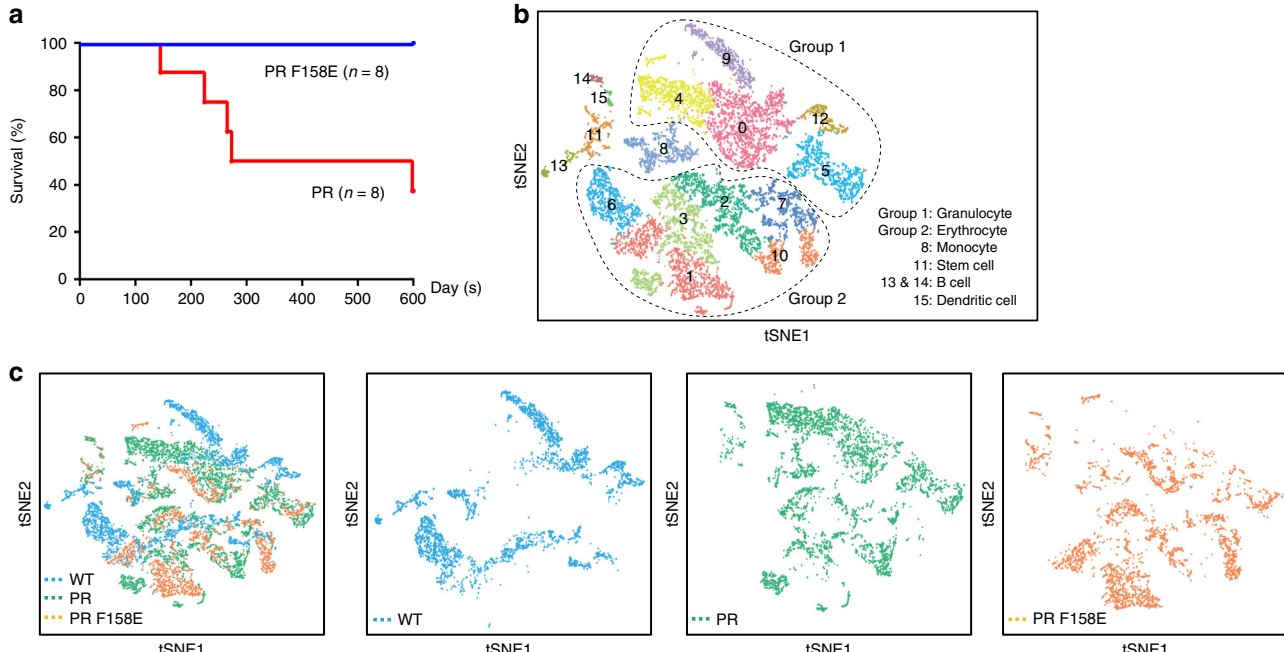

**Fig. 8** B1-box oligomerization is essential for PML-RARα and APL. **a** Survival data of PML-RARα (PR) or PML-RARα$_{F158E}$ (PR F158E) transgenic mice. The APL phenotypes of these transgenic mice are shown in Supplementary Fig. 8c–e. **b**, **c** Single-cell RNA sequencing analysis of B1-oligomerization in PML-RARα. The tSNE projections were calculated based on the genetic profiling of ~12,500 bone marrow cells from WT, PR, and PR F158E mice. Information about the sequencing runs, represented as raw bases and average reads/cell per sample, is included in the Supplementary Table 1. To avoid batch differences, the Seurat alignment method canonical correlation analysis (CCA)[62] was used for an integrated analysis of three datasets. For cell clustering, the graph-based method implemented in Seurat was used. In **b**, the cells/clusters were grouped into five different cell types, including granulocyte (clusters 0, 4, 5, 9, 12), erythrocyte (clusters 1, 2, 3, 6, 7,10), monocyte (cluster 8), stem cell (cluster 11), B cell (clusters13, 14), and dendritic cell (cluster 15). The tSNE plots of the key markers per cell type are shown in Supplementary Fig. 8a. In **c**, the WT, PR, and PR F158E cells are colored differently to highlight the altered PML-RARα-driven differentiation by B1 mutant. More information concerning the cell type and distribution percentages is shown in Supplementary Fig. 8bc

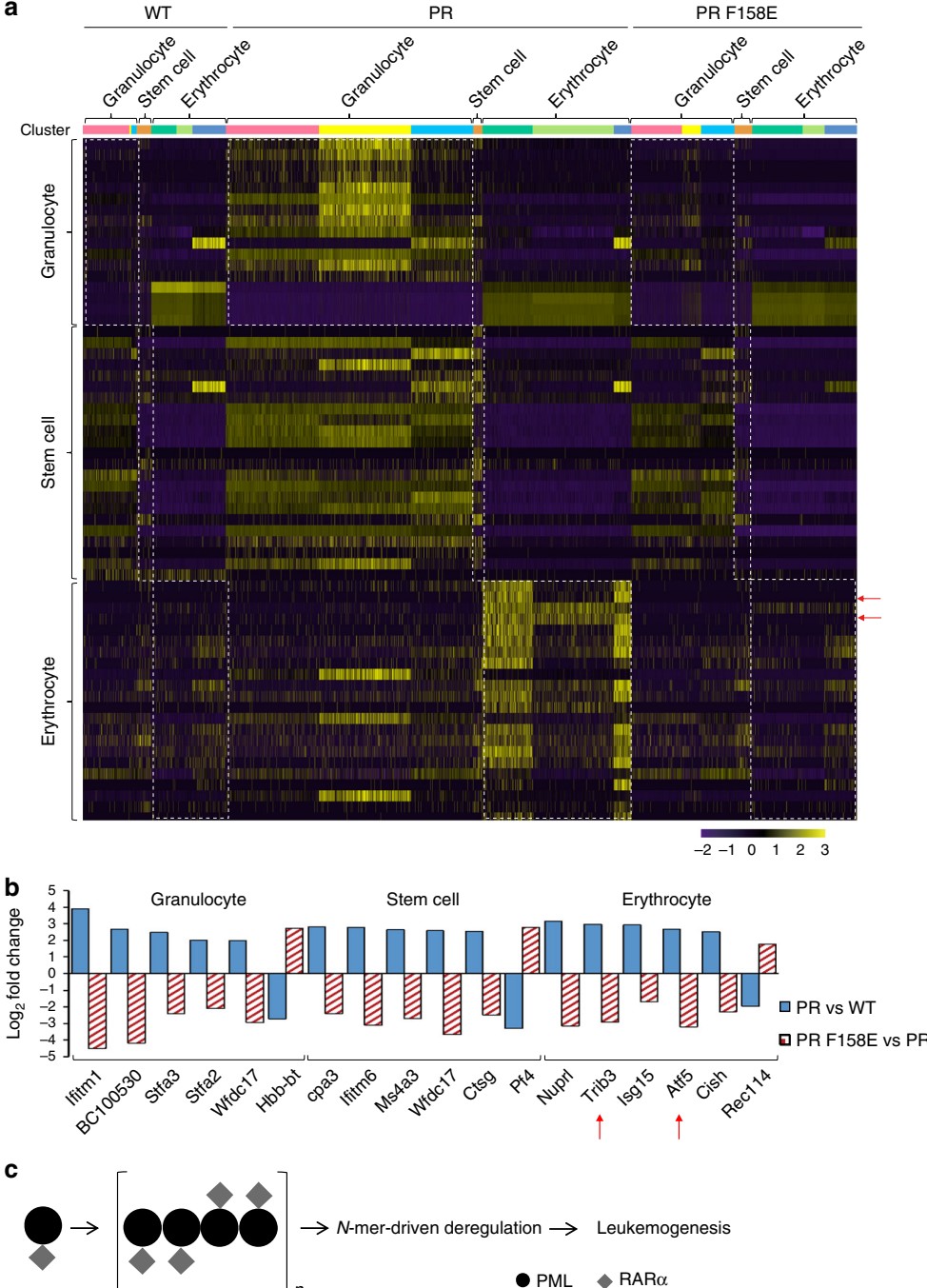

**Fig. 9** Abnormal transactivation regulated by B1 oligomerization. **a** Heat map of PML-RARα-oligomerization-driven transactivation. The equivalent sets of PML-RARα target genes in granulocyte, stem cells and erythrocyte cells are boxed, respectively. The transcription expression levels were calculated as normalized UMI value. The genes Trib3 and Atf5 recently reported in APL are highlighted with arrows[32]. **b** The differential expression analysis of PR and PR F158E datasets reveals a novel set of PML-RARα oligomerization target genes. The intersection/overlap between the up-regulation (PR vs. WT) and downregulation (PR vs. PR F158E) was used to predict PR-oligomerization-driven transactivation that might underpin APL leukemogenesis. Like **a**, Trib3 and Atf5 are highlighted with arrows. **c** A revised model of polymer-driven leukemogenesis

**Small angle X-ray scattering**. Different concentrations (i.e., 2, 4, and 8 mg/ml) of WT PML B1 and mutants were used for SAXS characterization. The X-ray scattering was recorded for a total $q$ range from 0.01 to 0.30 Å$^{-1}$ in beamline station BL19U2 (SSRF, China). The measurements were carried out with 1 s exposure time and repeated for 20 times to avoid radiation damage. Data subtraction and processing were performed using PRIMUS[29]. Crystal data fitting was done using the OLIGOMER algorism implemented in CRYSOL[29]. In the analysis of crystal fitting, the crystallographic B1 structures including monomer, dimers, tetramer, and N-mer were used to estimate the oligomeric distribution in solutions. For simplicity, the WT and mutant B1 oligomerizations at 2 mg/ml are shown in Fig. 4a–d. The

statistics of the data collection and scattering-derived parameters are shown in Supplementary Table 5.

**Oligomerization-dependent crosslinking**. For PML B1, the purified WT or mutant proteins (5 μg) were incubated with 0.05% (w/v), 0.1% (w/v), and 0.2% (w/v) glutaraldehyde (GA) on ice for 30 min, respectively. For HA-PML$_{1-256}$, similar cross-linking experiments were repeated using 0.3 μg purified WT or mutant proteins (0.3 μg). The total reaction volume was 20 μl. The buffer used in the cross-linking assay was 20 mM Hepes, pH 7.4. The cross-linking reactions were stopped by adding

**Table 1 Data collection and structure refinement statistics of PML B1-box**

| Protein | PML B1-box |
|---|---|
| *Data collection* | |
| Space group | $P2_1$ |
| Unit cell (Å, °) | |
| *a, b, c* | 36.7, 50.5, 51.4 |
| *β* | 101.4 |
| Molecule per ASU | 4 |
| Derivative | Native |
| Source/station[a] | BL17U |
| Wavelength (Å) | 1.282 |
| Resolution range (Å) | 50.4–2.06 |
| Observations ($I/\sigma(I) > 0$) | 45412 |
| Unique reflections ($I/\sigma(I) > 0$) | 11287 (1650) |
| High resolution shell (Å) | 2.17–2.06 |
| $R_{sym}$ (%)[b,c] | 17.7 (69.4) |
| $<I/\sigma(I)>$[c] | 5.5 (2.9) |
| Completeness[c] (%) | 98.0 (98.3) |
| Redundancy[c] | 4.0 (4.1) |
| $CC_{1/2}$ | 0.97 (0.61) |
| *Structure refinement* | |
| Resolution range (Å) | 25.2–2.06 |
| R-factor (%) | 18.6 |
| R-factor (high resolution shell)[d] | 24.6 |
| $R_{free}$ (%)[e] | 22.7 |
| $R_{free}$ (high resolution shell) | 27.7 |
| Total number of non-hydrogen atoms | 1700 |
| Protein atoms | 1609 |
| Water molecules | 82 |
| R.m.s. deviations[f] | |
| Bond length (Å) | 0.006 |
| Bond angle (°) | 0.993 |
| Wilson B-factor (Å$^2$) | 25.9 |
| Average B-factor protein atoms (Å$^2$) | 40.3 |
| Average B-factor water atoms (Å$^2$) | 45.5 |
| Ramachandran statistics (%) | |
| Most favored region | 97.4 |
| Allowed regions | 2.6 |
| Outliers | 0 |

[a]Beamline designations refer to the Shanghai Synchrotron Radiation Facility, Shanghai, P. R. of China
[b]$R_{sym} = S(I - \langle I \rangle)^2/SI^2$
[c]Overall, high resolution shell in parentheses
[d]High resolution shell: 2.2672–2.0600
[e]$R_{free}$ calculated using 5% of total reflections omitted from refinement
[f]R.m.s. deviations report root mean square deviations from ideal bond lengths/angles and of B-factors between bonded atoms[63]

1 µl 1 M Tris, pH 8.0, to the reaction mixture. The cross-linked PML B1 polymers were checked by an 8–16% gradient SDS-PAGE and coomassie blue staining. In the case of HA-PML$_{1–256}$, the cross-linked products were resolved by a 4–12% gradient SDS-PAGE and analyzed by immunoblot using anti-HA antibodies (Abcam).

**Circular dichroism.** In this experiment, the purified PML$_{1–256}$ and mutants were prepared by desalting the recombinant proteins and diluting them to 0.1 mg/ml using a buffer containing 10 mM CHES, pH 9.0. Far UV circular dichroism (CD) spectra were recorded from 185 to 260 nm at 20 °C with a time constant of 1 s using a Chirascan spectrometer (Applied Photophysics Ltd.). The spectra were the average of not less than three scans and presented as mean residue molar ellipticity [θ] (deg cm$^2$ dmol$^{-1}$). The secondary structure content was estimated using CD deconvolution program CDNN.

**Analytical ultracentrifuge.** The WT PML$_{1–256}$ or mutant eluants/peaks derived from GF were subjected to UC analysis. Sedimentation experiments were conducted using a Beckman XL-1 analytical ultracentrifuge equipped with an eight-cell rotor. The WT and mutant proteins kept in the buffer of 20 mM Tris, pH 8.0, 100 mM NaCl were spun at the speed of 130,000 g at 20 °C for > 12 h. The UC data were analyzed using SEDFIT, in which a continuous sedimentation coefficient distribution model was applied[61]. The sedimentation coefficient ($s = v/\omega^2 r$) was extrapolated to water at 20 °C.

**Mammalian two-hybrid assay.** This assay was performed using the CheckMate™ mammalian two-hybrid system (Promega) in 293T cells. For PML–PML interaction, the cDNAs of full-length WT PML isoform IV and mutants were engineered into pBIND and pACT vectors, respectively. For PML-partners interaction, the cDNAs of SUMOs/DAXX were cloned into pBIND vectors. The cDNAs of the WT and mutant PML isoform IV were engineered into pACT vectors. The 293T cells were then co-transfected with pG5/luc, pBIND-PML$_{WT}$/pBIND-SUMOs/pBIND-DAXX and pACT-PML$_{WT}$/PML$_{mutant}$ mixtures at 1:1:1 molar ratio using liposome Lipo2000 transfection protocol (Invitrogen). Twenty-four hours after transfection, the 293T cells were harvested. The relative luciferase activities were monitored by the Dual-Luciferase Reporter Assay System (Promega).

**PML NB biogenesis and PR speckle forming assays.** The WT PML isoform I (UniProt code, P29590–1), WT PML isoform IV (UniProt code, P29590-5), and their mutants were engineered into pEGFP-C1 vectors that contain an N-terminal GFP tag, respectively. The resulting GFP-PML and mutants were transfected into the homemade HeLa$^{Pml−/−}$ cells, in which the endogenous *pml* gene was knocked out by the CRISPR-Cas9 strategy. The cells were grown in Dulbecco's modified Eagle's medium containing 10% fetal bovine serum before they were subjected to immunofluorescence examination. The cells were fixed with 4% paraformaldehyde. The slides were examined with a Leica TCS SP8 or Zeiss LSM870 confocal fluorescent microscope. It has been reported that PR can oligomerize and form small speckles in nucleoli[30]. This was used in this report to check the presence of B1 oligomerization in PR. Similar to the NB formation assay described above, the WT PR and mutants were engineered into pEGFP-C1 vectors, followed by transfection into the HeLa$^{Pml−/−}$ cells. The expression and speckle formation activities were visualized by immunofluorescence.

**Immuno co-localization assay.** The PML partner proteins DAXX, SUMO2, and SUMO3 were engineered into pFLAG-CMV4 vectors, respectively. The FLAG-tagged DAXX, SUMO2, and SUMO3 were cotransfected into the HeLa$^{Pml−/−}$ cells with WT GFP-PML isoform I, WT GFP-PML isoform IV or their mutants at 1:1 molar ratio. Concerning arsenic treatment, the cells harboring the co-expression of PML and its partner protein were incubated with 2 µM As$_2$O$_3$ for 1 h. For visualization, the cells were fixed with 4% paraformaldehyde and incubated for 10 min in a solution of 0.25% Triton X-100 in PBS. Then incubated with primary antibody anti-Flag (Sigma-Aldrich) overnight at 4 °C. Alexa 568-labeled conjugated anti-mouse secondary antibody was used for detection. The slides were examined with a Leica TCS SP8 or Zeiss LSM870 confocal fluorescent microscope.

**PML sumoylation assay.** HA-PML isoform IV, HA-PML isoform I, and derivatives derived from pLVX-IRES-Puro vectors were co-transfected into HeLa$^{Pml−/−}$ cells with His-SUMO1/2/3. The cells were grown in Dulbecco's modified Eagle's medium containing 10% fetal bovine serum before western blot analysis. The sumoylation levels were detected using anti-HA antibody. Concerning arsenic response, the cells harboring WT PML and mutants were incubated with 2 µM As$_2$O$_3$ for 1 h before lysis using RIPA buffer containing 50 mM Tris-HCl, pH 8.0, 150 mM NaCl, 1% NP-40, 0.5% sodium deoxycholate, 1% PMSF, 1% protease inhibitor cocktail (Roch). The soluble and insoluble PMLs derived from sumoylation were fractioned by centrifugation (16,000g, 15 min). The pellets (i.e., the insoluble fraction) were solubilized by RIPA and 1% sodium dodecyl sulfate (SDS). Both the soluble and insoluble fractions were resolved in a 4–20% gradient SDS-PAGE and analyzed by immunoblot using anti-HA (Abcam ab9110, 1:5000 dilution) and anti-actin (Santa sc47778, 1:500 dilution) antibodies.

**Luciferase reporter assay.** The promoter regions harboring PU.1 motifs were cloned into the pGL3-basic luciferase reporter vector (Promega). The expression plasmids contained pLVX-IRES-PR, pLVX-IRES-PR W157E, pLVX-IRES-PR F158E and pCMV4-PU.1, respectively. The renilla luciferase plasmid pRL-SV40 (Promega) was used as control for transfection efficiency. The plasmids pGL-3-basic, pLVX-IRES-PR/mutants, pCMV4-PU.1 and pRL-SV40 were all co-transfected to 293T cells using Lipofectamine 2000 (Invitrogen). Twenty-four hours after transfection, cells were harvested for the determination of luciferase activities using the Dual-luciferase reporter assay kit (Promega).

**Transgenic mice.** PR and PML-RARα$_{F158E}$ (PR F158E) were expressed by human MRP8 promoter. All animal experiments and care were carried out as approved by the Animal Care and Use Committee at Experimental Animal Centre in Shanghai Jiao Tong University School of Medicine, China. Standard checks such as HE staining, flow cytometry analysis of c-Kit, Gr-1 and Mac-1 in spleen and bone marrow cells were used to confirm APL development.

**Single-cell RNA sequencing.** The normal FVB/N mice (termed WT) and the PR and PR F158E transgenic mice at the same age (i.e., 78 weeks) were used for single-cell RNA sequencing (scRNA-seq) analysis. All mouse experiments were conducted following the standard operating procedures approved by the Animal Care and Use Committee at Experimental Animal Centre in Shanghai Jiao Tong University School of Medicine, China. In brief, the bone marrow cells were firstly acquired

from PR and PR F158E mice, the former of which has displayed an early stage of hematological disorders as monitored/confirmed by fluorescence-activated cell sorting analysis and enlarged spleen size. For single cell suspension, the phosphate-buffered saline (PBS) buffer with 0.04% bovine serum albumin was used. The cell counts and vitality were then verified by TC20™ Automated Cell Counter (Thermo Fisher). Single cells were captured and barcoded in 10× Chromium Controller (10× Genomics). We used one mouse for each genotype. The single cell suspensions of PR and PR F158E were captured in one chip with two separated channels. The single cell suspension of WT was captured in another chip with one channel because of different dates for sample collection. Single cell RNA sequencing libraries were prepared using Chromium Single Cell 3'v2 Reagent Kit (10x Genomics). Sequencing libraries were loaded on an Illumina NovaSeq with 2 × 150 paired-end kits. The FASTQ files were analyzed with the Cell Ranger Software Suite (version 2.0; 10× Genomics). We compared the raw data with mouse genome reference (mm10) to generate the filtered gene-barcode matrix which contained valid cell barcodes and transcript UMI counts with Cell Ranger Count. The single cell RNA sequencing statistics are listed in Supplementary Table 1. The aggregated gene-barcode matrix of three mice model was normalized by Cell Ranger aggr based on depth. Then, we used the normalized aggregated gene-barcode matrix to perform preliminary analysis of scRNA-seq, in which we performed the $K$-means method to cluster cells. Clusters were visualized with t-distributed Stochastic Neighbor Embedding of the principal components (t-SNE) as implemented in Cell Ranger. The cell type of each cluster was manually determined by recognized marker genes.

To validate the accuracy of the results of Cell Ranger, we also utilized Seurat[62] to perform cell clustering. The filtered gene-barcode matrix of each mouse identified by Cell Ranger Count was inputted into Seurat. To further filter cells and genes in Seurat, we removed cell outliers (retain cells with 200–2500 genes), cells with high mitochondrial transcript proportion (>5%), as well as genes that were detected in less than three cells. The information of filtered cells and genes are listed in Supplementary Table 1. Then, we normalized the filtered gene-barcode matrix by total expression of each mouse. To avoid batch differences, we used the Seurat alignment method canonical correlation analysis (CCA)[62] for an integrated analysis of three datasets. The CCA method can exclude confounding variables that have effects on cells and identify the global transcriptional shifts among different datasets. We identified a shared correlation constructure of the datasets by CCA, then got the common variations among the three datasets. Gene dispersion analysis was used to select highly variable genes. Based on the first ten canonical correlation vectors, a new dimensional reduction was generated for further analysis. The clusters were visualized with t-SNE plots with resolution set 0.6 as implemented in Seurat. The cell type of each cluster was manually determined by recognized marker genes. Pair-comparisons (PR vs. WT, PR vs. PR F158E) were performed with stringent threshold value (log$_2$[fold change] å 1.5 and adjusted $p$ value/FDR < 0.05). In each cluster, we used the intersection/overlap between the upregulation (PR vs. WT) and downregulation (PR vs. PR F158E) to predict PR-oligomerization-driven transactivations that might underpin APL leukemogenesis (Supplementary Table 2).

**Reporting summary**. Further information on research design is available in the Nature Research Reporting Summary linked to this article.

## Data availability

Coordinates and structure factors have been deposited under accession code 6IMQ. The source data underlying Figs. 1c, d, 4a–c, 4e, f, 5a–c, 5e, f, 6b, 7a, c, and Supplementary Figs. 3b, 4a, 4c, 4e, 5b, 6b, and 7b are provided as a Source Data file. Other data are available from the corresponding author upon reasonable request.

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

## Acknowledgements

This work was supported by research grants 81770142, 81370620, 81570120, 31070645, 81800144, and 31800642 from National Natural Science Foundation of China, a research grant 20152504 from "Shanghai Municipal Education Commission—Gaofeng Clinical Medicine Grant Support", "The Program for Professor of Special Appointment (Eastern Scholar) at Shanghai Institute of Higher Learning", a research grant 11JC1407200 from SMSTC, a research grant 12ZZ109 from SME, Program for New Century Excellent Talents in University (NCET-10–9571), Samuel Waxman Cancer Research Foundation. We thank the staffs Shan Li, Wei Qi, Zhixin Wang from Novogene Bioinformatics Technology Co., Ltd., Beijing for assistance during single cell RNAseq analysis.

## Author contribution

Conceived and designed the experiments: G.M. Performed the experiments: Y.L., X.M., Z.M.C., H.W., P.W., W.W., N.C., L.Z. and H.Z. Analyzed the data: Y.L., X.M., Z.M.C., H.W., P.W., W.W., N.C., L.Z., H.Z., X.C., S.-J.C., Z.C. and G.M. Preparation of the figures: Y.L. and G.M. Wrote the paper: Y.L. and G.M. Supervised this work: G.M.

## Additional information

Competing interestThe authors declare no competing interests.

