## [Peer Review File · Nature Communications]

Reviewers' comments:

Reviewer #1 (Remarks to the Author):

Preamble

The promyelocytic leukemia protein forms nuclear bodies that have been ascribed many functions in the cell, including DNA repair, control of DNA tumour virus replication, and the control of cellular growth and differentiation in different tissue contexts and malignancies. The first interest in PML as a tumour suppressor comes from the study of the role of the PML-RARalpha fusion protein in acute promyelocytic leukemia, which is characterized by loss of PML nuclear bodies and is correlated with the expansion of promyelocytes and impaired hematopoietic differentiation. The formation of these bodies is mediated by intramolecular interactions between PML proteins via the RBCC domain (Ring Finger, B-box domains followed by a coiled-coil) and post-translational modification of PML isoforms (of which there are 6 nuclear forms) and partner proteins (e.g. DAXX, SP100, ATRX etc.) by sumoylation, which in turn promotes interactions between sumo-binding domains (SIMs) on PML and other body components and the sumo modification. These modes of body formation - i.e. a) via RBCC and b) via SUMO/SIM interactions - are not mutually exclusive but play overlapping roles in body formation and assembly of "functional" PML nuclear bodies that contain various proteins constitutively (such as DAXX/SP100) or under stress-conditions (such as p53). Recruitment/release of factors coupled with their post-translational modification at PML bodies, is thought to be the main function PML nuclear bodies in the cell. Thus, an integral part of how PML NB loss/disruption affects the cell is the potential loss of post-translational modifications (e.g. sumoylation, phosphorylation) of nuclear body components and their mislocalization in the cell.

In this manuscript the authors describe how novel mutants in the B1 box (e.g. W157E, F158E) of the RBCC domain affect PML multimerization, using X-ray crystallography, SAXS, sedimentation/gel filtration, and cell assays for nuclear body formation (in the presence and absence of arsenic). Finally, they extend their analyses to the effects of these mutations in B1 box on hematopoietic differentiation in the context of expression of the PML-RARalpha fusion that drives APL; concluding that the F158E mutation, ostensibly by blocking PML multimerization, prevents leukemogenesis by restoring differentiation during hematopoiesis.

It is well known that PML-RARalpha multimerization (and the RBCC domain) is required for the differentiation block seen in APL (reviewed in Jensen et al., 2001 Oncogene). This study provides more precise information on the role of the B1-box, which is of interest to biochemists and potentially those researchers developing drugs that block protein-protein interactions. Overall this is an interesting addition to the PML and RBCC domain canon of literature, provides a new mutational tool for future studies but would be improved by taking the analysis a little further, in addition to cleaning up the language and addressing some minor points (as outlined below).

Experiments and points to consider:

1) As mentioned in the preamble, PML body function is linked to the body components localized/sequestered and/or post-translationally modified at PML bodies. The authors have not evaluated the localization of key PML NB components, such as DAXX, SP100, ATRX and SUMO in PML -/- HeLa cells and relative to KO cells expressing PML WT and mutants (W157E, F158E and I122P/V123P) in cells alone as in Figure 4E and treated with arsenic as in Figure 5B. The results of these experiments have implications for the mechanisms by which body disruption by these mutants might affect cell function.

2) What are the functional consequences of loss of PML multimerization on transcription factors involved in hematopoietic differentiation or apoptosis? For example, what is happening to the protein modification and levels C/EBP β , C/EBP ϵ and PU.1, and are their transcriptional outputs affected? (see Weng XQ et al., 2016 Leukemia Research) ? Is p53 response to apoptotic stimuli impaired by B1-box mutants, which could affect survival of leukemia cells?

3) In all add-back experiments in the HeLa PML -/- cells (or hematopoietic cells as needed), at least two isoforms of PML should be used to control for possible isoform specific effects, particularly in the addressing points 1 and 2 above. Two commonly used isoforms are PML I (most abundant) and PML IV, which is in general more tumour suppressive and lacks exon 9 of PML I.

Minor points

- 1) The sumoylation sites should be indicated in Figure 1A depicting the PML protein. B-box B1 contains a SUMO-site at K160 and B2 contains a sumoylation site at K226.
- 2) The PML isoform being used in each figure (e.g. Figure 1) should be described in the figure legends as well as in the Methods and Materials. Is PML isoform IV being used for all in cellulo studies? What isoform is being used in Figures 1, 4 and 5
- 3) Molecular weight markers should be included for all PML western blots to indicate size of isoform being expressed and sumo-modified forms.
- 4) Figure 6 tSNE plots for WT, PR and F158E in panel C should be shown as separate panels for each condition for clarity, the overlapping image does not adequately show the different patterns. In addition PR is not defined, which the reviewer takes to mean PML-RARalpha expressing.
- 5) "WT", "PR" and P/R are not defined in the legend for Figure 7. What is wild type in this figure, no expression of PML-RARalpha fusion?
- 6) The document contains some typos and grammar errors that should be addressed in the next revision. e.g. on page 12 "Methods" is spelled wrongly as "METHEODS". These errors are found throughout the manuscript.
- 7) Finally, the authors could broaden discussion to comment on whether W157, F158 are hot or cold spots for mutation the PML gene in cancers. Is PML multimerization function required in some contexts to sustain cancer progression/development? This is interesting in the context of leukemias in general as PML function is required for maintaining a pool of leukemia-initiating (stem-like) cancer cells.

Reviewer #2 (Remarks to the Author):

The authors report production and characterisation of a sumoylated fragment of the PML nuclear body protein TRIM1. Historically, production of this protein has been challenging due to problems with solubility. One of the selling points of this paper stated in the introduction is production of "the largest RBCC fragment ever purified".

The paper begins by observing from SEC and AUC data that when purified, the large construct (composed of 3-4 domains) is prone to multimerisation. This observation concludes the characterisation of this "largest fragment". The focus of the paper then moves to structural characterisation of a single domain (B1) by X-ray crystallography and SAXS. The crystal contacts are described with the assertion that they represent a biologically relevant multimerisation mechanism. SAXS data is presented measured from the B1 domain construct that was crystallised. The data is somewhat featureless and is interpreted as representing a mixture of 75% monomer and 25% larger multimers taken from the crystal packing. I would suggest that the SAXS data probably does support the presence of multimers in the solution but whether these multimers, that represent a small proportion of the total sample, can be confidently identified as the assemblies seen in the crystal packing is not clear. The situation is further complicated by the possibility that the B1 domain may be somewhat flexible in solution, particularly the extended SD1 loop that does not seem to have many contacts with the more structured SD2 domain. Mobility of this region would be predicted to increase the radius of gyration and flatten out the Porod slope to some extent, an effect that may detract from the confidence with which it can be asserted that the crystal contacts are real.

Mutations made at the crystal contacts seem to have an effect on protein multimerisation in vitro though no mutant form resolves itself as a single species which possibly suggests that multimerisation is somewhat non-specific.

The results of a two-hybrid assay support the hypothesis that mutations at the crystal contacts may effect multimerisation in vivo. The additional in vivo microscopy, single cell sequencing/sorting work supports the idea that the B1 domain may be playing a role in driving multimerisation. However, I do not believe that the evidence presented is sufficiently compelling to support the inference that the crystal packing seen in the B1 domain structure is relevant to our understanding of the full length protein in the context of nuclear bodies.

Reviewer #3 (Remarks to the Author):

In this paper, Li et al. successfully purify the ProMyelocyticLeukemia (PML) protein, which is crucial to understand the biogenesis of PML nuclear bodies. They also provide the crystal structure of the B1-box (B1) subunit. The atomic resolution and the very detailed description about the B1 structure, will undoubtedly benefit future studies on the PML protein or other proteins containing the widely observed B1 subunit. Interestingly, the authors unravel a novel and sequential oligomerization mechanism driven by B1. This opens the possibility that other oligomers are formed by the same mechanism. Importantly, oligomerization is observed in many oncogenic fusions. The authors address the implication of PML-RAR in leukemogenesis in vivo and with state-of-the-art single-cell technologies. They identify the oligomerisation transcriptional signature that could promote leukemia development, thus opening the avenue for the exploration of new therapeutic targets. Altogether, the findings of this paper are relevant and make a significant contribution to the field. However, some issues need to be further clarified by addressing the following points and providing additional data.^[1]

Major points:

1) In the results section is mentioned: "The mutants W157E, F158E and I122P/V123P, although expressed normally in HeLapml^{-/-} cells, precluded NB formation (Figure 4ef)". However, data showing the protein expression for the different mutants is missing. Please include this data in a supplementary panel.

2) In the results section "B1-box multimerization in PML-RAR" the authors ask the question "whether B1-box multimerization is present in this oncogenic fusion". However, this question is not addressed (immunofluorescence, as for PML, is not provided). Instead, the authors assume that PML-RAR multimerizes in vivo and address whether PML-RAR drives leukemogenesis. Therefore, statements such as "we also observed a strict correlation between APL development and B1-oligomerization" should be re-phrased or stated milder.

Also regarding to the in vivo experiment, apart from the survival data they should demonstrate that mice suffered APL and hence provide additional information. Add a supplementary figure showing how APL was monitored and about the phenotype of the sacrificed mice (WBC count, splenomegaly? anemia?, FACS staining showing infiltration of leukemia cells in different organs: spleen, bone marrow and blood etc)

3) The authors perform single-cell RNA sequencing with state-of-the-art technology (10X Genomics) but only use Cell Ranger (10X software) for the analysis. This software is suitable for the first analysis steps (demultiplexing and mapping) and for a first exploratory analysis. However, in order to profit the massive and complex data generated by a single-cell RNA sequencing experiment, bioinformatics expertise is required, which I am afraid is missing here. Many R packages (Seurat, Monocle, Bioconductor tutorials etc) provide flexible pipelines that could have uncovered more findings.

Regarding the single-cell presented data, insufficient methodological detail is provided and these are the points to be clarified/corrected:

- In the results and methods section specify that is RNA sequencing, at least the first time. The term "single-cell sequencing" or "single-cell analysis" is vague and can refer to RNA, DNA, ATAC-seq etc
- Indicate how many mice were pooled per genotype. Pooling provides more robust findings.
- In the methods is not mentioned that WT mice were also used.
- Cell loading: indicate the single-cell capture strategy. Where all the cells captured in the same or in subsequent runs? How many chip channels were used per run and per genotype?
- No information at all about sequencing. Which sequencing-platform was used? How many sequencing runs? Average reads/cell per genotype etc
- Important to indicate how many single cells were captured per genotype (both in the methods and in Sup Fig4. Mention how many cells were initially captured and how many passed the quality

control and were used for downstream analysis. I would suggest to add a supplementary table including sequencing and quality metrics information.

- Specify the criteria and thresholds used for cell filtering (% of reads mapping to mitochondria? number of detected genes? number of UMI? etc).
- Sup Fig4: include a legend in the figure indicating the cell types.
- Important: the transcriptional expression levels should be normalized, above all, when sequenced data from different libraries is pooled. The statement: "The transcription expression levels were calculated as UMI values" is not acceptable. Please, normalize the data and indicate the normalization method.
- In the tSNE visualization, was any co-variate regressed out? (accounting for cell cycle, batch effect etc).
- Which clustering method was used? And how were the clusters annotated? Show in a supplementary figure some key markers per cluster (violin plot or tSNE plot coloured according to the expression level of specific genes). Despite APL-free condition, any cluster contains promyelocytes (eg: co-expression of Gr1 and CD34), and are they mainly found in P/R? A sub-clustering analysis could be performed to identify potential APL-initiating cells. At what age where the mice used?
- Fig 7a: for a nicer visualization, rotate the Y axis text 180 degrees and remove the empty X-axis ticks.
- In the results section "PML-RARaF158E displayed completely different single cell profiling" is a very vague description. Be more specific by indicating which cell types are differently distributed and provide a % comparison.
- In the results section, add a sentence explaining more specifically figure 7a. Indicate the expression pattern of target PML-RAR genes. And how were these genes identified? Was a differential expression analysis performed? Which fold change and/or significance level (FDR) was considered? In the abstract is said that "F158E significantly alters transcription" but any adjusted p-val/FDR is mentioned in the methods.

Minor points:

- 1) Although the manuscript is clearly and coherently written, there are still substantial typographical, spelling and grammatical errors. I strongly suggest that a native English speaker reads and corrects the paper. I would also avoid strange wordings such as "in celleo".
- 2) It is also important to check for consistencies throughout the paper. E.g: they both use PML-RAR or PML/RAR terms; P/R or PR (Figure 6) etc
- 3) Fig 1a: the text referring to the RBCC components is not readable.
- 4) SupFig1 title: "Sequence alignment of TRIM B1-boxes". Change it for B-boxes since both B1 and B2 boxes are shown
- 5) Fig 4d: define Fold Change in the figure legend. Is the plot normalized to PACT vector:pBind-PML interaction (=1) and the fold change calculated comparing all interactions to PACT vector:pBind-PML?
- 6) Fig 4f and S3bd indicate that a representative replicate is shown. However, to strengthen the finding, I suggest to include the results from the 6 replicates (violin or boxplot).
- 7) Typo: F158A instead of F157A in text sentence "Complementary to these results, we also observed the same results in W157K, W157A, F158K, F157A and D120-124 (Supplementary Figure 3ab)
- 8) The authors report an unprecedented B1-driven oligomerization mechanism in PML and PML-RAR (repeated many times). However, they do not explain it further. They should contextualize more this finding. For instance, in the discussion, briefly discuss what are the common oligomerization mechanism and why the one reported in the paper is special.
- 9) The authors claim (end of introduction) that "the in vitro and in vivo characterizations of PML...". However, PML oligomerization and nuclear body formation is not addressed in vivo (only one panel in figure 6 addresses the role of PML-RAR oncogenic fusion). Therefore, the in vivo statement should be removed unless in vivo data on PML oligomerization and nuclear body formation is provided.
- 10) In the discussion sentence "The mutation targeting B1-oligomerization failed to trigger APL in vivo", should be the opposite.
- 11) In the discussion sentence "the RNA-seq analysis showed a marked difference between PML-RARa and PML-RARaF158E" in terms of what? Otherwise is just a redundant sentence (same as

previous one).

Reviewer #4 (Remarks to the Author):

The manuscript by Li and coworkers describes the E. coli expression of two PML fragments; a 256 aa N-terminal fragment and the 48 aa B-box 1. The shorter construct was crystallized and the structure was refined at 2 Å resolution. The crystal contacts seen in the B-box1 structure were interpreted to be physiological relevant for the oligomerization behavior of PML. Residues involved in crystal contacts were mutated and mutants were analyzed in vitro and in vivo.

The nice thing about this comprehensive study is that it bridges the gap between pure biochemical/biophysical and animal studies. Unfortunately, the wealth of data comes with the lack of details. For many experiments described in this manuscript, detailed experimental information is missing. Just a few examples: Figure 1c shows the SEC analysis of the large fragment. Four peaks are labeled, but no information about the column material nor running buffer is given (e.g. page 13, lines 3-4: "... the S100 and Superdex 200 were selected ...", which of them was used for Fig. 1c?). Some reference values for column calibration would be helpful. What are the calculated molecular weights from figures 1c? Which sedimentation coefficient was plotted in figure 1d? If it is the experimental value, which buffer was used? Typically, the sedimentation coefficient is extrapolated to water at 20°C. What are the fits (curves and chi2 values) for fitting the SAX data in figure 4a with monomers or dimers? Generally, the chi2 value decreases when higher oligomers are included. Regarding SAX, the Kratky plot could be informative about inter-domain flexibility in solution. Furthermore, I could not find any reference to a sequence database entry (e.g. Uniprot) about which protein has actually been cloned and analyzed nor to any PDB entry where the coordinates, structure factors or perhaps frames have been deposited.

Reading the manuscript I get the impression that oligomerization is exclusively due to crystal contacts of the 48 aa fragment, although this is less than 1/10 of the whole PML protein. Do the authors believe that the RING, B2 and coiled coil domains are irrelevant for oligomerization? Although the presented PML B1 box is just a remote homologue of B-boxes shown in figure S1, there is many structural and oligomerization data on these B-boxes as well. The authors are kindly asked to compare their results with the surface epitopes of other B-boxes where structural data is available. Are equivalent residues of PML F158, W157, I122 and V123, known to be involved in the oligomerization of other TRIM proteins? The authors just refer to the NMR structure 2MVW (Ref. 26), where they disagree with the mutational analysis of this NMR structure. How similar are the crystal- and NMR structures of the PML B1 box?

The entire mutational analysis presented in this manuscript relies on the assumption that only the quaternary structure is altered but the tertiary structure remains intact. Is there any indication that this assumption is correct? V123P and I122P are structurally harsh mutations. I agree that it is the only way to challenge the formation of the beta-sheet, but what about V123A? Is it structurally neutral? Even if the tertiary structure of mutants W157E and F158E are unchanged, which is likely, the results show that these hydrophobic patches are involved in oligomerization. If the oligomers are structurally similar to the crystal contact seen in the B1 box structure is still an assumption. In principle, they could interact with any other hydrophobic surface patch. To confirm that the W157/F158 crystal contacts are physiologically relevant compensating mutations of the W157E and F158E mutants would be required.

The following is our point-to-point responses to reviewers' concerns:

REVIEWER 1

The promyelocytic leukemia protein forms nuclear bodies that have been ascribed many functions in the cell, including DNA repair, control of DNA tumour virus replication, and the control of cellular growth and differentiation in different tissue contexts and malignancies. The first interest in PML as a tumour suppressor comes from the study of the role of the PML-RARalpha fusion protein in acute promyelocytic leukemia, which is characterized by loss of PML nuclear bodies and is correlated with the expansion of promyelocytes and impaired hematopoietic differentiation. The formation of these bodies is mediated by intramolecular interactions between PML proteins via the RBCC domain (Ring Finger, B-box domains followed by a coiled-coil) and post-translational modification of PML isoforms (of which there are 6 nuclear forms) and partner proteins (e.g. DAXX, SP100, ATRX etc.) by sumoylation, which in turn promotes interactions between sumo-binding domains (SIMs) on PML and other body components and the sumo modification. These modes of body formation - i.e. a) via RBCC and b) via SUMO/SIM interactions - are not mutually exclusive but play overlapping roles in body formation and assembly of "functional" PML nuclear bodies that contain various proteins constitutively (such as DAXX/SP100) or under stress-conditions (such as p53). Recruitment/release of factors coupled with their post-translational modification at PML bodies, is thought to be the main function PML nuclear bodies in the cell. Thus, an integral part of how PML NB loss/disruption affects the cell is the potential loss of post-translational modifications (e.g. sumoylation, phosphorylation) of nuclear body components and their mislocalization in the cell.

Reply: We greatly appreciate this reviewer's insightful comments/suggestions, which have led to more vigorous investigations and better understandings of PML nuclear body assembly and its partners' recruitment (see more results below).

In this manuscript, the authors describe how novel mutants in the B1 box (e.g. W157E, F158E) of the RBCC domain affect PML multimerization, using X-ray crystallography, SAXS, sedimentation/gel filtration, and cell assays for nuclear body formation (in the presence and absence of arsenic). Finally, they extend their analyses to the effects of these mutations in B1 box on hematopoietic differentiation in the context of expression of the PML-RARalpha fusion that drives APL; concluding that the F158E mutation, ostensibly by blocking PML multimerization, prevents leukemogenesis by restoring differentiation during hematopoiesis. It is well known that PML-RARalpha multimerization (and the RBCC domain) is required for the differentiation block seen in APL (reviewed in Jensen et al., 2001 Oncogene). This study provides more precise information on the role of the B1-box, which is of interest to biochemists and potentially those researchers developing drugs that block protein-protein interactions. Overall, this is an interesting addition to the PML and RBCC domain canon of literature, providing a new mutational tool for future studies.

Reply: Again, we warmly thank the reviewer's appreciation in our work and findings.

1) As mentioned in the preamble, PML body function is linked to the body components localized/sequestered and/or post-translationally modified at PML bodies. The authors have not evaluated the localization of key PML NB components, such as DAXX, SP100, ATRX and SUMO in PML -/- HeLa cells and relative to KO cells expressing PML WT and mutants (W157E, F158E and I122P/V123P) in cells alone as in Figure 4E and treated with arsenic as in Figure 5B. The results of these experiments have implications for the mechanisms by which body disruption by these mutants.

Reply: As suggested, we have now examined the co-localization between PML and its partner proteins including DAXX, SP100 and SUMO2/3 in HeLa^{Pml^{-/-}} cells (Figure 1ab

below). In addition, the mutations of K160R (sumoylation site mutation) and L73E (RING tetramerization mutation) are also investigated in this assay. As expected, DAXX and SP100 can co-localize with WT PML nuclear bodies (NBs), but not with the K160R mutant. This is consistent with the previous recognition that sumoylation is critical for PML-partner interaction (Sahin et al., J. Cell Biol, 2014). In line with this, the B1-box and RING mutations (i.e. W157E, F158E, I122P/V123P and L73E) can also impair DAXX/SP100/SUMOs recruitment, highlighting our previous proposal that PML oligomerization might precede sumoylation during NB biogenesis. Of note, the I122P/V123P mutant displays residual NB formation activity, and as a result, the NB-enhanced partners recruitment can be observed (Figure 1ab below). In order to check whether oligomerization has a direct impact upon partners recruitment, the mammalian-two-hybrid assay is used. As shown in Figure 1c below, the B1 mutations W157E, F158E and I122P/V123P significantly weaken the interactions between PML and its partners. In addition, when the B1 mutants are subjected to arsenic trioxide (ATO) treatment, we could observe NB re-appearance, followed by NB-enhanced partners recruitment (Figure 1d below). All the related results concerning DAXX and SUMO2/3 can be found in the revised manuscript. Altogether, these results suggest that the RBCC oligomerization, which might precede PML sumoylation and partners recruitment, is likely to be the first step in PML NB biogenesis.

a**b**

Figure 1. PML RBCC oligomerization and partner proteins recruitment. a, b) Co-localizations of PML and partners. c) The mammalian-two-hybrid characterization of PML-partner interaction. d) Arsenic (ATO) treatment and its impact on PML NB formation and partners recruitment. The PML-DAXX/SUMO2/SUMO3 interacting results, together with the similar experiments repeated in PML isoform I, are now reported in the revised Figure 6 and Supplementary Figures 5-6. Concerning the figure labelling in this letter and the revised manuscript, “PML” means PML isoform IV and “PML_I” means PML isoform I.

2) What are the functional consequences of loss of PML multimerization on transcription factors involved in hematopoietic differentiation or apoptosis? For example, what is happening to the protein modification and levels C/EBP β , C/EBP ϵ and PU.1, and are their transcriptional outputs affected? (see Weng XQ et al., 2016 Leukemia Research)? Is p53 response to apoptotic stimuli impaired by B1-box mutants, which could affect survival of leukemia cells?

Reply: As suggested, we have used luciferase assay to monitor the impact of PML oligomerization upon transcription factors such as PU.1 (Figure 2a below). As reported (Wang et al, Cancer Cell, 2010), WT PML-RAR α (PR) can inhibit the transactivation activity of PU.1. We can observe the same result with WT PR (Figure 2a below). In striking contrast, this inhibitory effect is greatly alleviated by the oligomerization-targeting mutants W157E and F158E (Figure 2a below). Supportively, a similar result could be observed in apoptosis assay, in which the apoptotic OCI-AML3 cells with stable expression of PML and mutants were labelled with annexin-V/PI and detected by flow cytometry (Figure 2b below). This restates the significance of multimerization in PML function and cell survival. However, when concerning the PML-related transcription factors such as C/EBP β , C/EBP ϵ , PU.1 and p53, our preliminary study fails to identify an impact that PML oligomerization might have upon these TFs (Figure 2c below). In light of our new finding in single cell RNA sequencing analysis, we will advise colleagues/readers to invest more research efforts in the PR oligomerization target genes in the future study (revised Figure 9 and Supplementary Table 2).

Figure 2. Functional consequences of PML and PML-RAR α oligomerization. a) Luciferase assay. **b)** PML-driven apoptosis assay. WT PML and mutants were stably transfected into OCI-AML3 cells. After 48 hours, the cells were harvested. The apoptotic OCI-AML3 cells were labelled with annexin-V/PI and detected by flow cytometry. **c)** PML multimerization and its impact upon the expression levels of p53, C/EBP β , C/EBP ϵ , PU.1 in OCI-AML3/U937 cells. The results in Figure 2a are now reported in the revised Supplementary Figure 7b.

3) In all add-back experiments in the HeLa *PML*^{-/-} cells (or hematopoietic cells as needed), at least two isoforms of PML should be used to control for possible isoform specific effects, particularly in the addressing points 1 and 2 above. Two commonly used isoforms are PML I (most abundant) and PML IV, which is in general more tumour suppressive and lacks exon 9 of PML I.

Reply: As suggested, the experiments are now repeated in the PML isoform I (Figure 3 below). The results concerning PML isoform IV and PML isoform I are both reported in the revised manuscript.

Figure 3. More experimental observations in PML isoform I, PML_I. a) PML_I NB formation assay. b) PML_I auto-sumoylation assay. c) Co-localizations of PML_I and partner proteins. d) ATO treatment and its impact on PML_I NB formation and PML_I-partners interactions. These results are now reported in the revised Supplementary Figure 5.

Minor points

1) The sumoylation sites should be indicated in Figure 1A depicting the PML protein. B-box B1 contains a SUMO-site at K160 and B2 contains a sumoylation site at K226.

Reply: Only three sumoylation sites K65/K160/K490 are reported in PML. These SUMO-sites are now indicated in the revised Figure 1a.

2) The PML isoform being used in each figure (e.g. Figure 1) should be described in the figure legends as well as in the Methods and Materials. Is PML isoform IV being used for all in cellular studies? What isoform is being used in Figures 1, 4 and 5?

Reply: The PML isoforms used in this study are now clarified in the figure legends and the M&M section.

3) Molecular weight markers should be included for all PML western blots to indicate size of isoform being expressed and sumo-modified forms.

Reply: The molecular weight markers are now included in all western blots and SDS-PAGES.

4) Figure 6 tSNE plots for WT, PR and F158E in panel C should be shown as separate panels for each condition for clarity, the overlapping image does not adequately show the different patterns. In addition, PR is not defined, which the reviewer takes to mean PML-RAR α expressing.

Reply: The tSNE plots are now shown according to the reviewer's suggestion (revised Figure 8bc). PR means PML-RAR α . This is clarified in the revised manuscript.

5) "WT", "PR" and P/R are not defined in the legend for Figure 7. What is wild type in this figure, no expression of PML-RAR α fusion?

Reply: WT means normal FVB/N mice (i.e. without PML-RAR α expression). PR means PML-RAR α transgenic mice. P/R is a typo and has been removed in the revised manuscript.

6) The document contains some typos and grammar errors that should be addressed in the next revision. e.g. on page 12 "Methods" is spelled wrongly as "METHEODS". These errors are found throughout the manuscript.

Reply: As non-native English speakers, we are very grateful for the reviewers' kindness and patience with our English struggle. As requested, we have tried our best to correct the typo and grammar errors in the revised manuscript.

7) Finally, the authors could broaden discussion to comment on whether W157, F158 are hot or cold spots for mutation the PML gene in cancers. Is PML multimerization function required in some contexts to sustain cancer progression/development? This is interesting in the context of leukemias in general as PML function is required for maintaining a pool of leukemia-initiating (stem-like) cancer cells.

Reply: W157 and F158 mutations are not yet reported in PML-related cancer patients. In the previous study (Wang et al., Nat Commun., 2018), we have shown that RING oligomerization is also required for APL development, supportive of our discovery in PML B1. In addition, multimerization is often associated with cancer-related proteins beyond APL (Weiner et al., Oncogene, 2005; Pal K et al., Structure, 2017; Wu YM et al., Cancer discovery). More interestingly, similar W157- and F158-like hydrophobic interfaces are often observed in these oncogenic oligomers. Thanks to the reviewer's kind/interesting suggestion, we have broadened our discussion (see more discussion in Page 13).

REVIEWER 2

The authors report production and characterization of a sumoylated fragment of the PML nuclear body protein TRIM1. Historically, production of this protein has been challenging due to problems with solubility. One of the selling points of this paper stated in the introduction is production of “the largest RBCC fragment ever purified”.

Reply: We warmly thank the reviewer’s kind interest in our findings.

The paper begins by observing from SEC and AUC data that when purified, the large construct (composed of 3-4 domains) is prone to multimerisation. This observation concludes the characterization of this “largest fragment”. The focus of the paper then moves to structural characterization of a single domain (B1) by X-ray crystallography and SAXS. The crystal contacts are described with the assertion that they represent a biologically relevant multimerisation mechanism.

Reply: We understand the reviewer’s concern in crystallographic finding. More biophysical characterizations of PML B1, PML₁₋₂₅₆, and their mutants are carried out to validate the unexpected B1 oligomerization in solution (see more results below).

SAXS data is presented measured from the B1 domain construct that was crystallised. The data is somewhat featureless and is interpreted as representing a mixture of 75% monomer and 25% larger multimers taken from the crystal packing. I would suggest that the SAXS data probably does support the presence of multimers in the solution but whether these multimers, that represent a small proportion of the total sample, can be confidently identified as the assemblies seen in the crystal packing is not clear.

Reply: In order to find out whether the SAXS multimers are random aggregates, a structure-based SAXS approach is used (Figure 4a-d below). When the W157 and F158 positions are perturbed, the B1 dimerization is significantly stalked, precluding B1 tetramerization and higher-order assemblies (N-mer). Similar results can be observed in the crosslinking assay using PML B1 and PML₁₋₂₅₆ (Figure 4e below). When the crosslinker glutaraldehyde (GA) is incubated with the WT PML B1, we can observe the crosslinking effect as monitored by SDS-PAGE (Figure 4e below). In comparison, the mutants W157E, F158E, and I122P/V123P display noticeable resistance to GA-mediated crosslinking (Figure 4e below). Supportively, this is also the case for PML₁₋₂₅₆ and mutants (Figure 4f below). We think that these new findings, together with other results presented elsewhere in this report, help to define the novel B1 oligomerization in solution.

Figure 4. More *in vitro* evidences of B1 oligomerization. a-d) Structure-based SAXS characterizations. e, f) Crosslinking of PML B1, PML₁₋₂₅₆ and mutants by glutaraldehyde (GA). These results are now reported in the revised Figure 4.

The situation is further complicated by the possibility that the B1 domain may be somewhat flexible in solution, particularly the extended SD1 loop that does not seem to have many contacts with the more structured SD2 domain. Mobility of this region would be predicted to increase the radius of gyration and flatten out the Porod slope to some extent, an effect that may detract from the confidence with which it can be asserted that the crystal contacts are real.

Reply: The Kratky plot and the CD spectroscopy show that the recombinant proteins of PML B1, PML₁₋₂₅₆ and mutants are well folded and suitable for SAXS analysis (Figure 5 below). Concerning the crystal contact, we have used structure-based SAXS and crosslinking assay to check the various B1 dimeric interfaces (also see our reply above).

Figure 5. a) Kratky plots of WT B1 and mutants. b) CD spectroscopy of WT B1 and mutants. These results are now shown in the revised Supplementary Figure 3.

Mutations made at the crystal contacts seem to have an effect on protein multimerisation in vitro though no mutant form resolves itself as a single species which possibly suggests that multimerisation is somewhat non-specific.

Reply: As reported by ourselves and other labs (Wang et al., Nat Commun., 2018; Jensen et al., Oncogene, 2001; Fagioli et al., Oncogene, 1998; Fanelli et al., JBC, 2004), RING and coiled-coil (CC) domains can contribute to the overall intertwining of RBCC. Concerning the PML₁₋₂₅₆ mutants W157E, F158E and I122P/V123P, the RING and CC oligomerization might be responsible for the residual polymerization as observed in gel filtration and ultracentrifugation.

The results of a two-hybrid assay support the hypothesis that mutations at the crystal contacts may affect multimerisation in vivo. The additional in vivo microscopy, single cell sequencing/sorting work supports the idea that the B1 domain may be playing a role in driving multimerisation. However, I do not believe that the evidence presented is sufficiently compelling to support the inference that the crystal packing seen in the B1 domain structure is relevant to our understanding of the full-length protein in the context of nuclear bodies.

Reply: We understand the reviewer's concern and agree that the B1 oligomerization reported here is rather unexpected. Therefore, a wide range of biophysical, biochemical and cellular techniques are used for cross-validations (revised Figures 4-7 and Supplementary Figures 3-7). All the results are consistent, supportive of B1 oligomerization in PML and PML-RAR α . More importantly, this finding is echoed in the *in vivo* studies using PML-RAR α transgenic mice and single cell RNA sequencing technologies (revised Figures 8-9 and Supplementary Figure 8).

REVIEWER 3

In this paper, Li et al. successfully purify the ProMyelocyticLeukemia (PML) protein, which is crucial to understand the biogenesis of PML nuclear bodies. They also provide the crystal structure of the B1-box (B1) subunit. The atomic resolution and the very detailed description about the B1 structure, will undoubtedly benefit future studies on the PML protein or other proteins containing the widely observed B1 subunit. Interestingly, the authors unravel a novel and sequential oligomerization mechanism driven by B1. This opens the possibility that other oligomers are formed by the same mechanism. Importantly, oligomerization is observed in many oncogenic fusions. The authors address the implication of PML-RARalpha in leukemogenesis *in vivo* and with state-of-the-art single-cell technologies. They identify the oligomerisation transcriptional signature that could promote leukemia development, thus opening the avenue for the exploration of new therapeutic targets. Altogether, the findings of this paper are relevant and make a significant contribution to the field.

Reply: We are very grateful for the reviewer's kind interest in our work.

Major points:

1) In the results section is mentioned: "The mutants W157E, F158E and I122P/V123P, although expressed normally in PML $-/-$ HeLa cells, precluded NB formation (Figure 4ef)". However, data showing the protein expression for the different mutants is missing. Please include this data in a supplementary panel.

Reply: The protein expressions of WT PML and mutants are now included in the revised manuscript (Figure 6 below).

Figure 6. The expression levels of PML and mutants in HeLa^{Pml^{-/-}} cells. These results are now reported in Supplementary Figure 7a.

2) In the results section "B1-box multimerization in PML-RARalpha" the authors ask the question "whether B1-box multimerization is present in this oncogenic fusion". However, this question is not addressed (immunofluorescence, as for PML, is not provided). Instead, the authors assume that PML-RARalpha multimerizes *in vivo* and address whether PML-RARalpha drives leukemogenesis. Therefore, statements such as "we also observed a strict correlation between APL development and B1-oligomerization" should be re-phrased or stated milder.

Reply: As suggested, we have studied the B1 oligomerization in PML-RAR α (Figure 7 below). The result is in good agreement with those observed in PML. The speckle-forming ability of PML-RAR α is greatly impaired by the mutations of W157E, F158E and I122P/V123P. This, together with the *in vivo* data presented in the revised Figures 8-9 and Supplementary Figures 7-8, support the B1 oligomerization in PML-RAR α .

Figure 7. B1 oligomerization in PML-RAR α (green) as monitored by speckle-formation assay. This result is now reported in Supplementary Figure 7a.

Also, regarding to the in vivo experiment, apart from the survival data they should demonstrate that mice suffered APL and hence provide additional information. Add a supplementary figure showing how APL was monitored and about the phenotype of the sacrificed mice (WBC count, splenomegaly? anemia?, FACS staining showing infiltration of leukemia cells in different organs: spleen, bone marrow and blood etc.)

Reply: As requested, we also report the APL phenotypes of the WT, PR and PR F158E mice in the revised manuscript.

Figure 8. APL phenotypes in WT, PR and PR F158E transgenic mice. WT, the normal FVB/N mice (i.e. without PML-RAR α gene). These results are now reported in the revised Supplementary Figure 7c-e.

3) The authors perform single-cell RNA sequencing with state-of-the-art technology (10X Genomics) but only use CellRanger (10X software) for the analysis. This software is suitable for the first analysis steps (demultiplexing and mapping) and for a first exploratory analysis. However, in order to profit the massive and complex data generated by a single-cell RNA sequencing experiment, bioinformatics expertise is required, which I am afraid is missing here. Many R packages (Seurat, Monocle, Bioconductor tutorials etc.) provide flexible pipelines that could have uncovered more findings.

Reply: As requested, the single cell RNA sequencing data are now re-analyzed by the Seurat package (Figure 9 below).

Figure 9. A representative diagram/result of the single cell RNA sequencing analysis by Seurat package. All the new results are reported in the revised Figures 8-9 and Supplementary Figure 8.

Regarding the single-cell presented data, insufficient methodological detail is provided and these are the points to be clarified/corrected:

Reply: All the methodological details concerning the single cell RNA sequencing analysis are carefully updated in the Material & Method section.

- In the results and methods section specify that is RNA sequencing, at least the first time. The term “single-cell sequencing” or “single-cell analysis” is vague and can refer to RNA, DNA, ATAC-seq etc..

Reply: It is single cell RNA sequencing. This has been clarified in the revised manuscript.

- Indicate how many mice were pooled per genotype. Pooling provides more robust findings.

Reply: We used one mouse per genotype. Visible skin lesions, enlarged spleen size and FACS analysis of bone marrow cells were used to confirm the early APL development in the sacrificed PR mouse (Figure 10 below). The numbers of cells filtered by Seurat are 3,816 (WT), 4,806 (PR) and 3,874 (PR F158E).

Figure 10. Early APL-like phenotypes in the sacrificed PR mouse chosen for single cell RNA sequencing analysis.

- In the methods is not mentioned that WT mice were also used.

Reply: WT is referred to the normal FVB/N mice (i.e. without PML-RAR α). PR means the transgenic mice harboring PML-RAR α . PR F158E means the mutant mice harboring PML-RAR α _{F158E}.

- Cell loading: indicate the single-cell capture strategy. Where all the cells captured in the same or in subsequent runs? How many chip channels were used per run and per genotype?

Reply: Single cells are captured by Gel-Beads-in-Emulsion (GEM) strategy and barcoded in 10x Chromium Controller (10x Genomics). We use one mouse for each genotype. The single cell suspensions of PR and PR F158E are captured in one chip with two separated channels. The single cell suspension of WT is captured in another chip with one channel.

- No information at all about sequencing. Which sequencing-platform was used? How many sequencing runs? Average reads/cell per genotype etc.

Reply: Sequencing libraries are loaded onto an Illumina NovaSeq with 2×150 paired-end kits. Information about sequencing runs, which is represented as raw bases and average reads/cell per sample, is now included in the revised Supplementary Table 1.

- Important to indicate how many single cells were captured per genotype (both in the methods and in Sup Fig4. Mention how many cells were initially captured and how many passed the quality control and were used for downstream analysis. I would suggest to add a supplementary table including sequencing and quality metrics information.

Reply: All these information is now shown in the table below.

	WT	PR	PR F158E
Platform	Novaseq	Novaseq	Novaseq
Raw Bases(G)	220.88	227.3	263.21
Number of cells identified by Cell Ranger	4,161	5,588	4,578
Median reads per cell identified by Cell Ranger	176,941	135,581	191,641
Median genes per cell identified by Cell Ranger	917	1,167	824
The number of Cells filtered by Seurat	3,816	4,806	3,874

Table 1. Single cell RNA sequencing statistics. These results are now reported in the revised Supplementary Table 1.

- Specify the criteria and thresholds used for cell filtering (% of reads mapping to mitochondria? number of detected genes? number of UMI? etc.).

Reply: The filtered gene-barcode matrix of each mouse identified by Cell Ranger Count is inputted into Seurat. To further filter cells and genes in Seurat, we remove cell outliers (to retain cells with 200-2,500 genes), cells with high mitochondrial transcript proportion (>5%), as well as genes which were detected in less than three cells.

- Sup Fig4: include a legend in the figure indicating the cell types.

Reply: This is revised according to the reviewer's suggestion (revised Supplementary Figure 8bc).

- Important: the transcriptional expression levels should be normalized, above all, when

sequenced data from different libraries is pooled. The statement: “The transcription expression levels were calculated as UMI values” is not acceptable. Please, normalize the data and indicate the normalization method.

Reply: When aggregating data from different libraries, Cell Ranger “aggr” function normalizes sequencing depth by down-sampling the read counts across libraries. In Seurat, we normalize the filtered gene-barcode matrix by total expression of each mouse. The statement is now corrected as follows, “*The transcription expression levels were calculated as normalized UMI values*”.

• In the tSNE visualization, was any co-variate regressed out? (accounting for cell cycle, batch effect etc.).

Reply: To avoid batch differences, we use the Seurat alignment method canonical correlation analysis (CCA) (Butler et al., Nat Biotechnol, 2018) for an integrated analysis of three datasets. The CCA method can exclude confounding variables that have effects on cells and identify the global transcriptional shifts among different datasets. We identify the shared correlation constructor of the datasets by CCA and then get the common variations among the three datasets. Based on the first 10 canonical correlation vectors, a new dimensional reduction is generated for further analysis.

• Which clustering method was used? And how were the clusters annotated?

Reply: For Cell Ranger, we use a normalized aggregated gene-barcode matrix to perform a preliminary analysis of scRNA-seq, in which we perform the K-means method to cluster cells. For Seurat, we perform the graph-based method to cluster cells.

Show in a supplementary figure some key markers per cluster (violin plot or tSNE plot coloured according to the expression level of specific genes).

Reply: As suggested, the tSNE plots of the key markers per cluster are shown in the revised Supplementary Figure 8a.

Despite APL-free condition, any cluster contains promyelocytes (e.g.: co-expression of Gr1 and CD34), and are they mainly found in P/R? A sub-clustering analysis could be performed to identify potential APL-initiating cells. At what age where the mice used?

Reply: As suggested, we have checked the co-expression of Gr1 and CD34 among WT, PR and PR F158E (Figure 11 below). When comparing the stem cells between PR and PR F158E, Gr-1⁺/CD34⁺ cells can be exclusively observed in the PR mouse. However, a similar co-expression is also found in the WT mouse. Based on these results, we think it might be too early to think/recognize these cells as APL-initiating cells. Concerning the second question, the same 78-week-old mice are used in the single cell RNA sequencing analysis.

Figure 11. The co-expression of Gr1⁺/CD34⁺ cells (red) in the WT, PR and PR F158E mouse. The stem cell cluster is highlighted with circles.

• Fig 7a: for a nicer visualization, rotate the Y axis text 180 degrees and remove the empty X-axis ticks.

Reply: This is revised according to the reviewer's suggestion (revised Figure 9b).

• In the results section "PML-RARalphaF158E displayed completely different single cell profiling" is a very vague description. Be more specific by indicating which cell types are differently distributed and provide a % comparison.

Reply: As suggested, the information concerning the cell types and distribution percentages are now included in the revised manuscript (revised Supplementary Figure 8bc).

• In the results section, add a sentence explaining more specifically figure 7a. Indicate the expression pattern of target PML-RARalpha genes. And how were these genes identified? Was a differential expression analysis performed? Which fold change and/or significance level (FDR) was considered? In the abstract is said that "F158E significantly alters transcription" but any adjusted p-val/FDR is mentioned in the methods.

Reply: Pair-comparisons are performed with stringent threshold values: $\log_2[\text{fold change}] > 1.5$ and the adjusted p-value/FDR < 0.05 .

Minor points:

1) Although the manuscript is clearly and coherently written, there are still substantial typographical, spelling and grammatical errors. I strongly suggest that a native English speaker reads and corrects the paper. I would also avoid strange wordings such as "in cellego".

Reply: We appreciate reviewers' kindness and patience with our linguistic struggle. In the revised manuscript, we have tried our best to clean up the typo and grammatical errors.

2) It is also important to check for consistencies throughout the paper. E.g.: they both use PML-RARalpha or PML/RARalpha terms; P/R or PR (Figure 6) etc.

Reply: This is corrected.

3) Fig 1a: the text referring to the RBCC components is not readable.

Reply: Figure 1a is revised to make clear of the RBCC domain arrangement.

4) SupFig1 title: "Sequence alignment of TRIM B1-boxes". Change it for B-boxes since both B1 and B2 boxes are shown

Reply: This is corrected.

5) Fig 4d: define Fold Change in the figure legend. Is the plot normalized to PACT vector:pBind-PML interaction (=1) and the fold change calculated comparing all interactions to PACT vector:pBind-PML?

Reply: We can confirm that the interactions between WT and different mutants are all normalized against the pACT vector:pBIND-PML interaction (=1). This is clearly stated in the revised figure legend.

6) Fig 4f and S3bd indicate that a representative replicate is shown. However, to strengthen the finding, I suggest to include the results from the 6 replicates (violin or boxplot).

Reply: This is done according to the reviewer's suggestion.

7) Typo: F158A instead of F157A in text sentence "Complementary to these results, we also observed the same results in W157K, W157A, F158K, F157A and D120-124 (Supplementary Figure 3ab)

Reply: This is corrected.

8) The authors report an unprecedented B1-driven oligomerization mechanism in PML and

PML-RAR α (repeated many times). However, they do not explain it further. They should contextualize more this finding. For instance, in the discussion, briefly discuss what are the common oligomerization mechanism and why the one reported in the paper is special.

Reply: In previous studies, TRIM B-boxes are often found to mediate dimerization and trimerization, but not tetramerization or even higher-degree polymerization (N-mer). In this study, we report three different dimeric interfaces in PML B1. Concertedly, the W157-, F158- and SD1-interfaces can give rise to a remarkable net-like oligomerization. B1-mediated networking is not reported before. As suggested by reviewer (please also see our reply to reviewer 4 below), the PML B1 oligomerization is now compared against other oligomeric TRIM B-box structures. Thanks to the reviewers' kind suggestion, the structural superimpositions/comparisons help to identify the α 1 helix as a common oligomerization active site in TRIM B-boxes. All these have been carefully discussed in the section of "*versatile B-box oligomerization*" (Page 13).

9) The authors claim (end of introduction) that "the in vitro and in vivo characterizations of PML...". However, PML oligomerization and nuclear body formation is not addressed in vivo (only one panel in figure 6 addresses the role of PML-RAR α oncogenic fusion). Therefore, the in vivo statement should be removed unless in vivo data on PML oligomerization and nuclear body formation is provided.

Reply: As suggested, the PR-driven speckle formation assay is used to check the B1 oligomerization in PML-RAR α (revised Supplementary Figure 7a).

10) In the discussion sentence "The mutation targeting B1-oligomerization failed to trigger APL in vivo", should be the opposite.

Reply: This is corrected.

11) In the discussion sentence "the RNA-seq analysis showed a marked difference between PML-RAR α and PML-RAR α F158E" in terms of what? Otherwise is just a redundant sentence (same as previous one).

Reply: As suggested, we have rephrased text in the discussion section.

REVIEWER 4

The manuscript by Li and co-workers describes the *E. coli* expression of two PML fragments; a 256 aa N-terminal fragment and the 48 aa B-box 1. The shorter construct was crystallized and the structure was refined at 2 Å resolution. The crystal contacts seen in the B-box1 structure were interpreted to be physiological relevant for the oligomerization behavior of PML. Residues involved in crystal contacts were mutated and mutants were analyzed *in vitro* and *in vivo*.

The nice thing about this comprehensive study is that it bridges the gap between pure biochemical/biophysical and animal studies. Unfortunately, the wealth of data comes with the lack of details. For many experiments described in this manuscript, detailed experimental information is missing.

Reply: We warmly thank the reviewer's kind interest in our work. As requested, we have carefully updated the information/details in the figure legends and M&M section (also see below).

Just a few examples: Figure 1c shows the SEC analysis of the large fragment. Four peaks are labeled, but no information about the column material nor running buffer is given (e.g. page 13, lines 3-4: "... the S100 and Superdex 200 were selected ...", which of them was used for Fig. 1c?).

Reply: The column and buffer information is now provided in the figure legend and the M&M section. The result presented in Figure 1c is done with Superdex 200.

Some reference values for column calibration would be helpful. What are the calculated molecular weights from figures 1c?

Reply: The Superdex 200 column is calibrated by the standard molecular marker kit obtained from GE Healthcare (Figure 12 below). The molecular weights are now indicated in the revised Figure 1c.

Figure 12. Calibration of Superdex 200 used in this study.

Which sedimentation coefficient was plotted in figure 1d? If it is the experimental value, which buffer was used? Typically, the sedimentation coefficient is extrapolated to water at 20°C.

Reply: Sedimentation coefficients ($s=v/\omega^2r$) of Peaks 1/2/3 obtained from the SEC experiment are plotted in the revised Figure 1d. The buffer used in the ultracentrifugation experiment contains 20 mM Tris pH 8.0, 100 mM NaCl. In addition, we can confirm that the sedimentation coefficients are extrapolated to water at 20°C.

What are the fits (curves and chi2 values) for fitting the SAXS data in figure 4a with monomers or dimers? Generally, the chi2 value decreases when higher oligomers are included. Regarding SAXS, the Kratky plot could be informative about inter-domain flexibility in solution.

Reply: The crystal fitting of WT SAXS data using monomers or dimers are shown in Figure 13 below. In comparison to the good fitting by oligomeric algorithm ($\text{Chi}^2=1.29$, shown in the revised Figure 4), the Chi^2 values with monomers-only and dimers-only are much bigger (76.4 and 10.9, respectively). This, together with the structure-based SAXS and crosslinking characterizations, supports the claim of B1 oligomerization in solution. In addition, the Kratky analysis suggests the WT/mutants PML B1 proteins are well folded and suitable for SAXS characterization (also see our reply to reviewer 2).

Figure 13. More SAXS analysis. a) Kratky plots of WT PML B1 at the concentrations of 2, 4, 8 mg/ml. b) Crystal fitting of PML B1 monomers-only and dimers-only.

Furthermore, I could not find any reference to a sequence database entry (e.g. UniProt) about which protein has actually been cloned and analyzed nor to any PDB entry where the coordinates, structure factors or perhaps frames have been deposited.

Reply: The UniProt entry codes for PML isoforms I and IV are P29590-1 and P29590-5, respectively. The PDB code of the B1 structure reported here is 6IMQ. These have been clarified in the revised manuscript.

Reading the manuscript, I get the impression that oligomerization is exclusively due to crystal contacts of the 48 aa fragment, although this is less than 1/10 of the whole PML protein. Do the authors believe that the RING, B2 and coiled coil domains are irrelevant for oligomerization?

Reply: In previous studies, it has been shown that RING and coiled-coil (CC) oligomerizations are essential to PML NB formation (Wang et al., Nat Commun., 2018; Jensen et al., Oncogene, 2001; Fagioli et al., Oncogene, 1998; Fanelli et al., JBC, 2004). In this report, we demonstrate that B1 oligomerization is also critical for NBs. In particular, the observations that PML₁₋₂₅₆ (i.e. RBCC) mutants can undergo polymerization in gel filtration and ultracentrifugation suggest a cooperative oligomerization mechanism among RING, B-box and CC. In addition, we also use ATO response/treatment to investigate whether RING tetramerization might happen before B1 oligomerization. The lack of ATO rescue effect on L73E also (revised Figure 7) seems to imply a cooperative mechanism among RBCC, in which RING tetramerization might precede B1 polymerization in PML NB biogenesis.

Although the presented PML B1 box is just a remote homologue of B-boxes shown in figure S1, there is many structural and oligomerization data on these B-boxes as well. The authors are kindly asked to compare their results with the surface epitopes of other B-boxes where structural data is available. Are equivalent residues of PML F158, W157, I122 and V123, known to be involved in the oligomerization of other TRIM proteins?

Reply: We are grateful to the reviewer's kind suggestion. The structure alignments between PML B1-box and TRIM B-boxes (that display oligomeric activities) help to identify the $\alpha 1$ helix as an oligomeric hot spot in TRIM proteins (Figure 14 below).

Figure 14. Structural comparisons between PML B1-box and other oligomeric TRIM B-boxes. a) Structural superimposition between PML (green) and TRIM5 α (blue). RMSD = 1.5 Å. b-f) the conserved hydrophobic interactions in different dimeric/trimeric TRIM B-boxes. The bulky residues in the $\alpha 1$ helix are shown in stick representations. The conserved $\alpha 1$ helix is labelled. These results are now reported in the revised Supplementary Figure 2b.

The authors just refer to the NMR structure 2MVW (Ref. 26), where they disagree with the mutational analysis of this NMR structure. How similar are the crystal and NMR structures of the PML B1 box?

Reply: The structural superimposition between crystallographic and NMR B1s is showed in Figure 15 below. The SD2 sub-domain is highly conserved (RMSD = 1.2 Å). Of note, both SD1s remain relatively isolated from the SD2s, giving rise to a tea-cup-shape architecture. The conserved SD1-SD2 orientation/architecture is consistent with the Kratky plots mentioned above, in which no major inter-domain flexibility can be observed.

Figure 15. Structure comparison between NMR (salmon) and crystallographic (green) PML B1-box monomer.

The entire mutational analysis presented in this manuscript relies on the assumption that only the quaternary structure is altered but the tertiary structure remains intact. Is there any indication that this assumption is correct?

Reply: As monitored by CD spectroscopy and Kratky analysis, the mutations do not cause major structural alternation in PML B1 and PML₁₋₂₅₆ (revised Supplementary Figure 3ab). Consistently, the full-length WT PML and mutants share similar expression levels and stability in HeLa^{Pml^{-/-}} cells (revised Supplementary Figure 4a), further supporting the claim that B1 mutants do not cause major structural alternations in PML proteins.

V123P and I122P are structurally harsh mutations. I agree that it is the only way to challenge the formation of the beta-sheet, but what about V123A? Is it structurally neutral?

Reply: The results of I122A, I122P, V123A and V123P are shown below (Figure 16 below). Consistent with the I122P/V123P result, I122P and V123P both have a disruptive impact upon PML NB biogenesis, resulting in an increased level of diffused PML (Figure 16 below). In comparison, I122A and V123A are structurally neutral, causing little damages.

Figure 16. More structure-based characterizations of SD1-interface. These results are now reported in Supplementary Figure 4bc.

Even if the tertiary structure of mutants W157E and F158E are unchanged, which is likely, the results show that these hydrophobic patches are involved in oligomerization. If the oligomers are structurally similar to the crystal contact seen in the B1 box structure is still an assumption. In principle, they could interact with any other hydrophobic surface patch. To confirm that the W157/F158 crystal contacts are physiologically relevant compensating mutations of the W157E and F158E mutants would be required.

Reply: The complementary surfaces in macro-molecular interaction/assembly are often stemmed from a long history of protein evolution. For this reason, we think that the suggestion of using/engineering a compensating mutation/surface to rescue the damage created by W157E and F158E might not be feasible. Instead, we have checked the complementary surface, i.e. the A154 position 4.2 Å away from F158. As shown in Figure 17 below, A154E mutant consistently abrogates NB formation. This, together with other structure-based mutations presented in this study, helps to define a novel B1 oligomerization in PML.

Figure 17. More evidence for B1 oligomerization. a) A154 (black) is located 4.2 Å away from F158. b) A154E abrogates NB formation.

REVIEWERS' COMMENTS:

Reviewer #1 (Remarks to the Author):

The authors have done an admirable job in addressing all reviewer's concerns. Congratulations on an excellent study.

Reviewer #2 (Remarks to the Author):

My initial concerns over this manuscript related to the inference that the crystal packing of the B1 domain represents a biological multimerisation phenomenon. I recognise that this kind of non-stoichiometric multimerisation is somewhat challenging to study from a structural point of view. To address these concerns the authors have added a more detailed analysis of the SAXS data of wt and mutant B1 revealing differences in the multimerisation state. The result is further validated by a cross-linking experiment. I am satisfied that the authors have made a very reasonable attempt to rigorously test their hypotheses regarding multimerisation and believe that the work is now of sufficient quality for publication.

Reviewer #3 (Remarks to the Author):

Dear authors,

I have carefully checked the revised manuscript. I appreciate your considerable amount of work in order to address all the comments and improve the manuscript.

I am satisfied with the changes in regard to my previous concerns. Therefore, I support the publication of this manuscript in Nature Communications.

Of note, minor comments 4 and 7 have not been corrected.

I would like to add another minor comment on Figure 8: substitute "is now included in" by just "is included in". The last sentence refers to Sup Fig 8 and not 9 (Sup Fig 9 does not exist).

Kind regards,
Lucia

Reviewer #4 (Remarks to the Author):

My request were all handled adequately.